# Quantifying the missing link between forest albedo and productivity in the boreal zone

Aarne Hovi[1], Jingjing Liang[2], Lauri Korhonen[3], Hideki Kobayashi[4], Miina Rautiainen[1,5]

[1]Department of Built Environment, School of Engineering, Aalto University, P.O.Box 15800, 00076 AALTO, Finland

[2]School of Natural Resources, West Virginia University, P.O.Box 6125, Morgantown, WV 26505, USA

[3]School of Forest Sciences, University of Eastern Finland, P.O.Box 111, 80101 Joensuu, Finland

[4]Department of Environmental Geochemical Cycle Research, Japan Agency for Marine-Earth Science and Technology, 3173-25, Showa-machi, Kanazawa-ku, Yokohama, 236-0001, Japan

[5]Department of Radio Science and Engineering, School of Electrical Engineering, Aalto University, P.O. Box 13 000, 00076 AALTO, Finland

*Correspondence to*: Aarne Hovi (aarne.hovi@aalto.fi)

**Abstract.** Albedo and fraction of absorbed photosynthetically active radiation (FAPAR) determine the shortwave radiation balance and productivity of forests. Currently, the physical link between forest albedo and productivity is poorly understood, yet it is crucial for designing optimal forest management strategies for mitigating climate change. We investigated the relationships between boreal forest structure, albedo and FAPAR using radiative transfer model FRT and extensive forest inventory data sets ranging from southern boreal forests to the northern tree line in Finland and Alaska (N = 1086 plots). The forests in the study areas vary widely in structure, species composition, and human interference, from intensively managed in Finland to natural growth in Alaska. We show that FAPAR of tree canopies ($FAPAR_{CAN}$) and albedo are tightly linked in boreal coniferous forests, but the relationship is weaker if the forest has broadleaved admixture, or if canopies have low leaf area and the composition of forest floor varies. Furthermore, the functional shape of the relationship between albedo and $FAPAR_{CAN}$ depends on the angular distribution of incoming solar irradiance. We also show that forest floor can contribute to over 50% of albedo or total ecosystem FAPAR. Based on our simulations, forest albedos can vary notably across the biome. Because of larger proportion of broadleaved trees, the studied plots in Alaska had higher albedo (0.141–0.184) than those in Finland (0.136–0.171) even though the albedo of pure coniferous forests was lower in Alaska. Our results reveal that variation in solar angle will need to be accounted for when evaluating climate effects of forest management in different latitudes. Furthermore, increasing the proportion of broadleaved trees in coniferous forests is the most important means of maximizing albedo without compromising productivity: based on our findings the potential of controlling forest density (i.e., basal area) to increase albedo may be limited compared to the effect of favoring broadleaved species.

Keywords: FAPAR, conifer, broadleaved, radiative transfer, basal area, leaf area index, AGB, thinning

# 1   Introduction

Forest management practices, such as thinning and logging, alter the spatial, structural, and species composition of forests. Through an altered albedo and productivity, these management practices may cause profound impacts on climate. Because forest structure and species composition influence albedo, managing forests to increase albedo is a potential means of maximizing the climate cooling effects of forests (Bright et al., 2014; Alkama & Cescatti, 2016; Naudts et al., 2016). However, if forest management practices are altered in order to maximize albedo, productivity may be compromised, which would result in reduced carbon uptake as well as reduced timber production and corresponding economic losses. There is an urgent need to understand how forest management practices change forest albedo, and how forest albedo and productivity are interconnected.

Being the world's largest land-based biome, the boreal forest zone consists of vast forest areas under various human interference levels, from natural growth to intense silvicultural management. The biome plays an important role in controlling the global carbon and energy balances. It is estimated that the boreal forests comprise 32% of the total carbon in the world's forests, and account for a significant portion of the carbon uptake (Pan et al., 2011). In addition, the albedo of boreal forests varies considerably by forest structure, phenology, and snow cover (e.g., Ni & Woodcock, 2000; Kuusinen et al., 2012; Bright et al., 2013; Kuusinen et al., 2016).

Previous studies based on local in situ measurements, or remote sensing data for local to regional study areas have shown that boreal forest albedo is influenced by tree species, with broadleaved species rendering higher albedos than coniferous (Lukeš et al., 2013a, Kuusinen et al., 2014). Albedo of open areas or that of the forest floor is usually higher than in the canopy areas (Bright et al., 2014, Kuusinen et al., 2014), except for burned sites (Amiro et al., 2006). A declining trend in albedo with forest height or age has been observed for coniferous forests (Amiro et al., 2006; Kirschbaum et al., 2011; Bright et al., 2013; Kuusinen et al., 2016) and may be at least partly explained by the increasing leaf area index (LAI) and thus reduced contribution of the forest floor on albedo as the forests mature. Similarly, a declining trend in albedo with canopy density has been observed (Lukeš et al., 2013a).

Gross primary productivity of vegetation can be approximated by FAPAR, i.e. the fraction of PAR radiation (400–700 nm) absorbed by the vegetation canopy (Gobron & Verstraete, 2009), because photosynthesis is ultimately driven by the available solar energy. FAPAR is useful in monitoring and comparing productivity both spatially and temporally, especially in the absence of accurate growth and yield models, although it should be noted that productivity is affected also by light use efficiency (LUE) i.e. the efficiency by which plants convert the solar energy into photosynthesis products (Monteith, 1972). The main determinants of forest canopy FAPAR are leaf area index (LAI) and the directionality of incoming solar radiation (Majasalmi et al., 2014), because they determine the fraction of PAR radiation interceptable by the canopy. Similarly to

albedo, boreal forest FAPAR may differ by tree species (Roujean et al., 1999; Steinberg et al., 2006; Chasmer et al., 2008;
Serbin et al., 2013; Majasalmi et al., 2015) and stand age (Serbin et al., 2013), as both species and age are likely to influence
the LAI of the canopy.

Estimation methods set limits for the information that can be obtained on the spatial and temporal variation of albedo and
FAPAR. In situ measurements are accurate and can be directly linked with field measured forest structure. On the other
hand, they are extremely tedious and cannot cover large variations in forest structure. Satellite data provide ample coverage
of varying forest structures and wide spatial extent but may compromise spatial resolution and detail in the characterization
of forest structure. In addition, neither local albedo measurements nor satellite-based albedo products can explain the
causality between small-scale environmental management scenarios and changes in albedo or FAPAR. Radiative transfer
models offer a solution to these problems: forest radiative transfer models are a powerful tool for linking quantitative
changes in vegetation structure to albedo or FAPAR for large geographical regions. The models are parameterized using
mathematical descriptions of canopy structure (e.g., LAI, tree height, crown dimensions, stand density), optical properties of
foliage and forest floor, and spectral and angular properties of incoming radiation. Using these models, the albedo and
FAPAR of a forest can be calculated from readily measurable variables such as forest structure and leaf optical properties.

To our knowledge only one study has examined the relation between forest albedo and FAPAR (Lukeš et al., 2016). In that
study, coarse resolution satellite products (MODIS) were used and one geographical area (Finland) was studied.
Furthermore, previous studies on forest structure and albedo have mainly focused on local geographical scales (e.g. Finland,
Norway, but see Kuusinen et al. (2013) for comparison between Finland and Canada). Comparison of the relationships
between forest structure, albedo and FAPAR has not been performed across the biome, i.e. including both European and
North American boreal forests which have very different natural structures and forest management scenarios. Due to the
large north-south gradient and consequent structural diversity of forests in the boreal zone, the impact of forest management
on albedo cannot be expected to be the same.

Here we report results from quantifying the links between boreal forest structure, albedo and FAPAR ranging from southern
boreal forests to the northern tree line using detailed, large forest inventory data sets from Finland and Alaska (N = 1086
plots). The forests in the study areas vary widely in structure, species composition, and human interference, from intensively
managed (regularly thinned) forests in Finland to natural growth in Alaska. Using a radiative transfer modeling approach, we
quantify the effects of forest structure and species composition on albedo and FAPAR in order to answer how forest
management practices can be optimized for climate change mitigation. The significant benefit of the modeling approach is
that it enables to study structurally varying forests over large geographical areas, without compromising detail in the forest
structure representation or in the spatial resolution. Our study is therefore the first intercontinental study connecting albedo
and productivity of boreal forests, using accurate ground reference data.

## 2    Materials and methods

### 2.1    Study areas and field plots

This study is based on 1086 field plots located in Alaska, USA, and in Finland, between Northern latitudes of 60° and 68°. At these latitudes, solar zenith angle (SZA) at solar noon at midsummer ranges from 37° to 45°, and the annual average from 69° to 72°.

The field plots in Alaska (N = 584) were permanent sample plots established as part of Co-operative Alaska Forest Inventory that aims at long-term monitoring of forest conditions and dynamics (Malone et al., 2009). The plots were scattered in interior and southcentral Alaska across a region of about $300\,000\ \mathrm{km^2}$, from Fairbanks in the north to the Kenai Peninsula in the south (Fig. 1, for more details see Liang et al. (2015)). Some of the plots were measured more than once. We used only the most recent measurement of each plot. The plots in Finland (N = 502) were temporary or permanent sample plots. They were located at four separate sites: Hyytiälä (Majasalmi et al., 2015), Koli, Sodankylä, and Suonenjoki (Korhonen, 2011) ranging from southern to northern Finland (Fig. 1). Species-level attributes, including the number of stems per hectare, basal area, mean diameter at breast height, tree height, and length of living crown, were available for the plots. Basal area, the total cross-sectional area of stemwood ($\mathrm{m^2\ ha^{-1}}$) at breast height (i.e. at 1.3 m or 1.37 m), is a common measure of stand density in forest inventories and, combined with information on tree height, used as an indicator of need for silvicultural thinning operations.

Tree species in the Alaskan data were coniferous black spruce (*Picea mariana* (Mill.) B. S. P.) and white spruce (*Picea glauca* (Moench) Voss), and broadleaved quaking aspen (*Populus tremuloides* Michx.), black cottonwood or balsam poplar (*Populus trichocarpa* Torr. & Gray, *P. balsamifera* L.), Alaskan birch (*Betula neoalaskana* Sarg.), and Kenai birch (*Betula kenaica* W.H. Evans). Tree species in the Finnish data were coniferous Scots pine (*Pinus sylvestris* L.) and Norway spruce (*Picea abies* (L.) H. Karst), and broadleaved species comprising mainly of silver and downy birch (*Betula pendula* Roth, *B. pubescens* Ehrh.). The birches accounted for 89% of the basal area of the broadleaved species in Finland. The forest variables in the study plots are shown in Table 1, for all plots and separately for plots dominated by one species. The Alaskan and Finnish forests differed in structure. The forests in Alaska were on average denser in terms of basal area (Fig. 2), and contained larger proportion of broadleaved species than the Finnish forests (Table 1). Managed forests in Finland, which our plots mainly represent, are normally thinned 1–3 times during the rotation period so that coniferous species are favored. In our plots from Alaska, on the other hand, no thinnings were applied.

The plots in Finland were classified into six site fertility classes in the field, according to a local site type classification system (Cajander, 1949). We re-classified the original number of six fertility classes into three: "xeric", "mesic", and "herb-rich". The cover of grasses is highest in the herb-rich, and decreases towards the xeric type. The cover of lichens, on the

other hand, increases towards the xeric type (Hotanen et al., 2013). In the Alaskan plots no site fertility estimate was
available but the cover of each species in the forest floor had been estimated. We labeled the plots as lichen- or grass
dominated if either the cover of lichens or the total cover of herbs, grasses, rush, sedges, and fern was over 50%. The
remaining plots were dominated by shrubs and mosses or were a mixture of all species groups. Hereafter we refer to these
forest floor types as "grass", "shrub/moss", and "lichen". Forest floor types did not differ notably between forests dominated
by different tree species, except for Scots pine forests in Finland, which were often found in the xeric type and were almost
nonexistent in the herb-rich type (Table 2).
**2.2    Albedo and FAPAR simulations**
**2.2.1    Simulation model**
We simulated albedo and FAPAR using a radiative transfer model called Forest Reflectance and Transmittance model FRT.
It was originally published by Nilson and Peterson (1991) and later modified by Kuusk and Nilson (2000). FRT is a hybrid
type model that combines geometric-optical and radiative transfer based sub-models for modeling the first- and higher-order
scattering components, respectively. The model has been intercompared and validated within RAdiative transfer Model
Intercomparison exercise (RAMI) several times, including validation of both reflected and transmitted fractions of radiation.
The results from these tests are publicly available online (Joint Research Centre, 2016) and reported in peer-reviewed
scientific papers (e.g., Widlowski et al., 2007). In this study, we used a version of FRT modified by Mõttus et al. (2007). The
advantage of FRT is that it can be parameterized using standard forest inventory data, utilizing the allometric relations of
forest variables to foliage biomass and crown dimensions. This was important because field measurements of biophysical
variables (e.g., LAI) are not commonly available, as was the case also in our study plots.

FRT simulates stand-level bidirectional reflectance and transmittance factors (BRF, BTF) of a forest at specified
wavelengths. A 12×12 Gauss-Legendre cubature was used to integrate the simulated BRF and BTF values over the upper
and lower hemispheres, respectively. This resulted in upward scattered and downwelling (directly transmitted or downward
scattered) fractions of incoming radiation. The former is observed on top of, and the latter below the tree canopy. These
fractions were then used to calculate the shortwave broadband albedo and FAPAR. The simulations were carried out at 5 nm
resolution, and the albedo simulations covered a spectral region of 400−2100 nm which corresponds to the region from
which input data was available (see Section 2.2.2). The wavelengths below 400 nm account for 8%, and wavelengths over
2100 nm account for 2% of the solar irradiance on top of the atmosphere (Thuillier et al., 2003).

The shortwave albedo was obtained as a weighted sum of the spectral albedos, i.e. upward scattered fractions of incoming
radiation ( $f_l$ - ):

$$albedo = \sum_{\lambda=400}^{2100} w_\lambda \times f_\lambda^\uparrow , \tag{1}$$

The canopy and total FAPAR (FAPAR$_{CAN}$, FAPAR$_{TOT}$) were obtained as weighted sums of canopy absorption ($a_\lambda^C$) and total absorption ($a_\lambda^T$) over the PAR region:

$$FAPAR_{CAN} = \sum_{\lambda=400}^{700} w_\lambda \times a_\lambda^C , \tag{2}$$

$$FAPAR_{TOT} = \sum_{\lambda=400}^{700} w_\lambda \times a_\lambda^T , \tag{3}$$

The weights ($w_\lambda$) were obtained from the solar irradiance spectrum. Solar irradiance values (W m$^{-2}$) were scaled by dividing them with the total solar irradiance within the spectral region used (i.e., 400–2100 or 400–700 nm). The weights were thus unitless and summed up to unity. The canopy and total absorptions needed for FAPAR determination were obtained using upward scattered ($f_\lambda^\uparrow$) and downwelling ($f_\lambda^\downarrow$) fractions of incoming radiation, and the reflectance factor of the forest floor ($r_G$) as follows:

$$a_\lambda^C = 1 - f_\lambda^\uparrow - f_\lambda^\downarrow + r_\lambda^G \times f_\lambda^\downarrow , \tag{4}$$

$$a_\lambda^T = 1 - f_\lambda^\uparrow , \tag{5}$$

FAPAR$_{TOT}$ and FAPAR$_{CAN}$ were calculated separately, because the former is a measure of total ecosystem productivity whereas the latter is more closely linked with timber production. Our FAPAR$_{CAN}$ and FAPAR$_{TOT}$ do not separate green biomass from woody or dead branches or from litter on the ground, and the values therefore represent upper limits of available solar energy for photosynthesis in tree canopy, and in the ecosystem as a whole. Green biomass could not be separated, because no measurements on fraction of branch area to leaf area were made in the study plots. The same applies to the cover of litter on the forest floor which was available for some of the field plots but not for all of them. It should also be noted that open soils are rarely seen in boreal forests where the floor is covered by (at least) green mosses.

The simulations were carried out assuming direct illumination only ("black-sky") and completely isotropic diffuse
illumination ("white-sky"). Black sky albedo is not dependent on assumptions of atmospheric scattering properties, and is
commonly used as input in climate modeling (Schaaf et al., 2009). The white-sky case was included in order to represent a
realistic diffuse illumination scenario, i.e. cloudy days. The black-sky albedo and FAPAR were simulated for five SZAs
typical for the study areas: 40°, 50°, 60°, 70°, and 80°. We use terms "small SZA" and "large SZA" to refer to SZAs of 40°–
50° and 70°–80°, respectively.
In both black- and white-sky simulations, we used a top-of-atmosphere irradiance spectrum (Thuillier et al., 2003) as
weights, because the focus was on analysing the effects of forest structure, and we wanted to avoid introducing any
differences between the study areas due to imperfect parameterization of the atmosphere. However, in order to demonstrate
what would be the effect of atmosphere on our results, we applied a simple solar spectral model (Bird and Riordan, 1986) for
generating direct and diffuse components of at-ground solar irradiance spectrum. The direct and diffuse components were
then used to weight the spectral fluxes ( $f_l{}^-$ , $f_l{}^-$ ) simulated under direct and diffuse illumination, respectively. The
simulated actual (blue-) sky albedo and FAPAR were highly correlated (r >= 0.98) with black-sky ones, but blue-sky albedo
was higher than black-sky albedo when SZA was 70° or 80°. This is because scattering in the atmosphere increases as
function of SZA. Atmosphere scatters visible more effectively than infrared wavelengths, shifting the irradiance distribution
of incoming solar radiation towards longer wavelengths in which vegetation is more reflective. Because of high correlation
between black- and blue-sky results, we conclude that inclusion of atmosphere in the calculations would not significantly
change our conclusions, although would increase the simulated albedo values at large SZAs.

### 214 2.2.2    Model parameters

Tree crowns are represented in the FRT model by geometric primitives (cylinders, cones, ellipsoids, or combinations of
them). The foliage within a crown is assumed to be homogeneously distributed. The area volume density (area per unit
crown volume) of the foliage depends on the crown dimensions and on the foliage area per tree. Several tree classes can be
defined to represent different tree species or size classes. We used one class for each tree species but did not model size
variation within-species. In theory, a forest with trees of very different sizes would have a higher canopy surface roughness,
which could in turn lead to somewhat lower reflectance (albedo) values (Davidson and Wang, 2004). There were no field
measurements made on tree size distribution in our data from Finland, and we wanted to maintain the same calculation
procedure for both study areas, in order not to introduce any differences due to data processing steps. Because the maximum
number of species was seven in the Alaskan data, there was a maximum of seven tree classes per plot. We assumed ellipsoid
crown shape. The effect of crown shape on simulated forest BRDF was quantified in Rautiainen et al. (2004) who showed
that increasing the crown volume may either increase or decrease the simulated reflectance values, depending on canopy
closure. Ellipsoid has been shown to estimate crown volume accurately (Rautiainen et al., 2008) and was therefore used in
our study. Crown length was obtained from field measurements, and the crown radius was modeled using species-specific
allometric equations that require stem diameter as independent variable (Jakobsons, 1970; Bragg, 2001). Leaf dry biomass
was estimated with species-specific biomass equations (Repola, 2008; Repola, 2009; Yarie et al., 2007) and converted into
hemisurface i.e. half of total leaf area, using leaf mass per area (LMA) values from literature (Table 3). The performance of
wide range of crown radius and foliage mass models in forming the input of FRT has been reported by Lang et al. (2007).
The models used in our study were chosen based on geographical proximity to our study areas, and also on model
availability, particularly for the Alaskan species for which there existed a limited number of models. A slightly regular
spatial distribution pattern of trees was assumed, i.e. a value of 1.2 for the tree distribution parameter (a value of 1 indicates
Poisson distribution, Nilson, 1999). Other structural parameters needed in FRT simulations are presented in Table 3.

Optical properties i.e. reflectance and transmittance of the leaves and needles were obtained from laboratory spectrometer
measurements. The data for Finnish species were from Hyytiälä, Finland (Lukeš et al., 2013b). Spectra of birch were used
for all broadleaved species. The data for Alaskan species were from Superior National Forest, Minnesota, USA (Hall et al.,
1996). Data for all species could not be found separately, and therefore spectra of black spruce were used for both black and
white spruce, spectra of paper birch (*Betula papyrifera* Marsh.) were used for both birch species, and spectra of quaking
aspen were used for both quaking aspen and for the black cottonwood/balsam poplar group. Reflectance spectra of black and
white spruce needles have been found to be similar at least in the visible and near-infrared wavelengths (Richardson et al.,
2003). In our data, the spectra of coniferous species did not differ notably from each other (Fig. 3a). The same applied to
broadleaved species. Bark spectra for spruces and *Populus* sp. in Alaska were obtained from Hall et al. (1996), and for Scots
pine and Norway spruce in Finland from Lang et al. (2002) (Fig. 3b). Spectra of birch from Lang et al. (2002) were used for
birches in Alaska and for broadleaved species in Finland.

We used the annual shoot as a basic scattering element for conifers, similarly as in Lukeš et al. (2013a). This accounts for the
multiple scattering within shoot which results in the shoot albedo being lower than needle albedo. Shoot reflectance and
transmittance spectra were obtained by upscaling the needle single scattering albedo to shoot albedo (Rautiainen et al.,
2012), assuming that the reflectance to transmittance ratio of a shoot is equal to that of a needle. Bi-Lambertian scattering
properties of the scattering elements (leaves or shoots) were assumed.

Optical properties of the forest floor, i.e. reflectance factors at nadir view were obtained from field spectrometer
measurements. The data were collected from Poker Flat Research Range Black Spruce Forest, Alaska (measurements
described in Yang et al. (2014)), and from Hyytiälä, Finland (using similar methodology as in Rautiainen et al. (2011)).
Separate spectra for each forest floor type was used (Fig. 3c), because characteristics of the forest floor may influence the
forest reflectance and therefore also albedo (Rautiainen et al., 2007). Forest floor composition was assumed to be
independent of overstory density. Taking into account this dependence would have required quantitative data on forest floor
composition and spectral data on all of the forest floor components, which were not available. Analysis of a subset of plots
that had measurements of vegetation cover in the forest floor revealed that the cover of green vegetation in the forest floor
was only weakly correlated with the canopy closure of the overstory (Alaska r = -0.27; Hyytiälä (Finland) r = -0.33).

## 2.3    Data analyses


### 2.3.1    Albedo, FAPAR, and forest structure

We analyzed albedo and FAPAR ($FAPAR_{CAN}$, $FAPAR_{TOT}$) against each other, and against the forest variables. The analyses
were performed separately for Alaskan and Finnish data, and repeated for all simulated solar illumination conditions.
Because of the strong emphasis on forest management, main focus of the analysis was on tree species and tree height which
are usually measured as part of forest inventories. In addition, we analyzed albedo and FAPAR against effective leaf area
index ($LAI_{eff}$) and above ground biomass (AGB). $LAI_{eff}$ is calculated by FRT, and corresponds to the LAI of a horizontally
homogeneous, optically turbid canopy that has exactly the same transmittance (gap probability) as the canopy under
examination. AGB was calculated with individual-tree allometric equations (Repola, 2008; Repola, 2009; Yarie et al., 2007),
similarly as the foliage biomass.

In the next phase, all simulations were repeated assuming black soil (i.e., a totally absorbing background), in order to better
explain the dependencies of albedo on tree height and illumination conditions as well as to explain the differences of albedo
between Alaskan and Finnish forests. The albedo obtained in black soil simulation represents the plain canopy albedo
without the contribution of forest floor vegetation. We refer to this as "canopy contribution". Correspondingly, the
contribution of forest floor can be calculated by subtracting the canopy contribution from the albedo obtained when
assuming a vegetated forest floor. We refer to this as "forest floor contribution". Canopy and forest floor contributions can
be expressed as absolute values or relative values which sum up to 100%. For comparison with the results regarding albedo,
the forest floor contribution to total ecosystem FAPAR was also calculated, by subtracting $FAPAR_{CAN}$ from $FAPAR_{TOT}$.

We report the relationships of albedo and FAPAR against forest structure in Sect. 3.1. Results of these experiments are
needed for understanding the relations between albedo and FAPAR, which we report in Sect. 3.2.

### 2.3.2    Relative importance of density and tree species

To examine the relative importance of density and species composition, we analyzed albedo and $FAPAR_{CAN}$ against basal
area and the proportion of broadleaved trees. The analyses were performed separately for Alaska and Finland, and repeated
for all simulated solar illumination conditions. We excluded all plots with tree height less than 10 m from the analyses in
order to evaluate the effect of basal area independent of tree height. This was done based on the following reasoning. Basal
area was correlated with tree height when studying all plots (r = 0.61 (Alaska), r = 0.64 (Finland)). Preliminary analysis was
performed by successively removing plots with smallest trees and each time checking the correlation between height and
basal area. The correlation was reduced until a height threshold of 10 m (r = 0.40 (Alaska), r = 0.34 (Finland)) (cf. Fig. 2).
Therefore, the 10 m threshold was used to exclude the smallest trees from our analyses. Analysis of albedo and FAPAR
against basal area in this restricted set of plots gives an approximation of how thinnings would affect albedo and $FAPAR_{CAN}$
although in reality thinning a stand affects not only the basal area but also the spatial pattern and size distribution of trees.

Mean and standard deviation (SD) of albedo and $FAPAR_{CAN}$ in conifer-dominated forests were calculated for ten equally
spaced classes with respect to basal area. The center of the lowest class corresponded to the 5th and that of the highest class
to the 95th percentile of basal area in the data. To examine the effect of broadleaved proportion, mean and SD of albedo and
$FAPAR_{CAN}$ were calculated for ten equally spaced classes with respect to proportion of broadleaved trees, i.e. the
broadleaved proportions ranging from 0–10% to 90–100%. The analysis was repeated for sparse (basal area percentiles from
0th to 30th) and dense forest (basal area percentiles from 70th to 100th). We hypothesized that the proportion of broadleaved
trees would have smaller effect on albedo in sparse than in dense forest, because the forest floor has more significant role in
the sparse canopies. Results regarding the analysis of basal area and proportion broadleaved trees are reported in Sect 3.3.
**3    Results**
**3.1    Albedo, FAPAR, and forest structure**
Mean albedo of study plots in Alaska (0.141–0.184) was higher than in Finland (0.136–0.171). In general, the albedo of
broadleaved species was 42–130% higher than that of coniferous (Table 4). However, albedo varied greatly even among
coniferous species: in Alaska, the albedo of black spruce was 19–33% higher than that of white spruce, and in Finland, the
albedo of Scots pine forests was 20–31% higher than that of Norway spruce. Overall, the mean albedo of coniferous species
was 28–32% higher in Finland (0.131–0.161) than in Alaska (0.102–0.122). The mean albedos of broadleaved species in
Alaska did not differ significantly from each other (p > 0.05 in ANOVA), except in the white-sky case. Therefore the
broadleaved species were treated as one group hereafter. Increasing the SZA increased the black-sky albedos of all species
(Table 4).

The forest canopies in Alaska absorbed more PAR radiation than in Finland: mean $FAPAR_{CAN}$ in Alaska was 0.71–0.92 and
in Finland 0.63–0.89. At the smallest SZA (40°) in black-sky simulations, $FAPAR_{CAN}$ was highest for broadleaved species in
Alaska, followed by Norway spruce in Finland, white spruce in Alaska, and broadleaved in Finland (Table 4). Scots pine in
Finland and black spruce in Alaska had lowest $FAPAR_{CAN}$ among the species. The mean $FAPAR_{CAN}$ of broadleaved species
in Alaska did not differ significantly from each other in any of the simulated illumination conditions (p > 0.05 in ANOVA).
Increasing the SZA increased $FAPAR_{CAN}$ of all species and also reduced the differences between species. The relative
increase was smaller for broadleaved than for coniferous species. Therefore, the order of species in $FAPAR_{CAN}$ was different
at small and large SZAs (Table 4). $FAPAR_{TOT}$, an approximation of total ecosystem productivity, ranged from 0.93 to 0.98
and did not depend strongly on direction of illumination. $FAPAR_{TOT}$ of coniferous forests was higher than that of
broadleaved but the differences were not large in relative terms because $FAPAR_{TOT}$ was consistently high.

White-sky albedo corresponded best with black-sky albedo observed at SZA of 60° (r = 0.97, RMSE = 0.011, mean
difference = -0.001). It correlated strongly also with black-sky albedos observed at other SZAs (r ≥ 0.93). White-sky
$FAPAR_{CAN}$ corresponded best with black-sky $FAPAR_{CAN}$ observed at SZA of 40° (r = 1.00, RMSE = 0.04, mean difference
= 0.03) and very closely also with those observed at SZAs of 50° and 60°. On the other hand, it deviated notably from the
black-sky $FAPAR_{CAN}$ observed at SZAs of 70° and 80°. Because white-sky albedo and FAPAR were highly correlated with
their black-sky counterparts observed at small to moderate SZAs, we report the results hereafter for black-sky conditions
only, except for contribution of forest floor (Table 5) that is presented also for white-sky case, in order to maintain
comparability with results presented in Table 4.

Albedo decreased with increasing tree height in coniferous forests (Fig. 4). The decrease was most rapid at small tree heights
and saturated after the height reached approximately 10 m. When SZA increased, the difference in albedo between short and
tall forests became smaller (compare Fig. 4a,b to Fig. 4c,d). The albedo of broadleaved forests was similar for all tree heights
at the smallest SZA (40°). At large SZAs, however, there was an initial rapid increase in albedo for broadleaved forests with
small trees (Fig. 4d), after which the albedo remained stable. AGB was correlated with tree height (r = 0.72–0.78) and the
albedo responded to AGB with a similar saturating trend as in the case of tree height (Fig. 4e,f).

$FAPAR_{CAN}$ initially increased with increasing tree height, but saturated at large tree heights (Fig. 5). The saturation was
reached earlier and the maximum level of $FAPAR_{CAN}$ was higher at large SZAs. Similar saturating trends and SZA
dependencies were observed also against AGB although there was less variation in the y direction (Fig. 5e,f). $FAPAR_{TOT}$
increased as function of tree height in coniferous forests, and was stable in broadleaved forests (Fig. 6). However, the
variation in $FAPAR_{TOT}$ with tree height was small (values ranging from 0.93 to 0.98).

The average contribution of forest floor to total forest albedo depended on tree species and ranged from 4% to 53% (Table
5). It was largest at small SZAs and for tree species that had low $LAI_{eff}$ (see $LAI_{eff}$ values in Table 1). Forest floor
contribution decreased as a function of tree height (Fig. 7). The relation was even tighter when the forest floor contribution
was analyzed against $LAI_{eff}$ (not shown). This is logical because $LAI_{eff}$ is more directly linked with canopy transmittance
than is tree height. Increasing the SZA increased the canopy contribution in all plots. This caused the albedo to increase as a
function of SZA. Only a few sparse canopies (low $LAI_{eff}$) were an exception. In these plots, an increase in SZA reduced the
forest floor contribution more than it increased the canopy contribution. Results regarding contribution of forest floor to total
ecosystem FAPAR were similar as those observed for albedo, i.e. there were differences between tree species and decreasing
trends with increasing SZA (Table 5).

The differences in albedos between coniferous species, i.e. black spruce vs. white spruce, and Scots pine vs. Norway spruce,
were almost non-existent when comparing albedos obtained in black soil simulations (Table 5). This indicates that at least
some of the differences in albedos between coniferous species are explained by the varying forest floor contribution between
species. However, the differences in albedos between coniferous forests of Finland and Alaska remained, indicating that
other factors than forest floor influenced the species differences between the study areas.

$FAPAR_{CAN}$ varied notably more than albedo when comparing forests of same height, particularly at small SZAs (Fig. 4, Fig.
5). This can be explained by the link of $FAPAR_{CAN}$ with canopy interception. Interception was tightly related with $LAI_{eff}$ (not
shown), and it determined $FAPAR_{CAN}$ almost directly, because the foliage absorbed strongly at PAR wavelengths (Fig. 3a)
and therefore the multiple scattering was negligible. $LAI_{eff}$, in turn, varied considerably between forests of same height. The
outliers (tall trees, low $FAPAR_{CAN}$) in Fig. 5d were plots that had only few trees and therefore very low $LAI_{eff}$. Similarly,
Scots pine had lower $FAPAR_{CAN}$ compared to other species with same height (Fig. 5d). Further examination revealed that
Scots pine had short crowns and therefore low $LAI_{eff}$, although the leaf area per unit crown volume did not differ from the
other coniferous species. The strong link between $FAPAR_{CAN}$ and $LAI_{eff}$ explained also the observed species- and SZA
dependencies of $FAPAR_{CAN}$. At the lowest SZA (40°) the species-specific $FAPAR_{CAN}$ (Table 4) was strongly correlated with
species-specific $LAI_{eff}$ (Table 1) (r = 0.93). At large SZAs the canopy interception approached 100% at almost all $LAI_{eff}$
values (cf. Fig. 5c,d) and $FAPAR_{CAN}$ was therefore mainly determined by the absorption of the foliage at PAR wavelengths.
Leaves of broadleaved trees absorbed less than conifer needles, which explains why $FAPAR_{CAN}$ of broadleaved species did
not increase as rapidly as a function of SZA as did $FAPAR_{CAN}$ of coniferous species (Table 4).
**3.2     Relation of albedo to FAPAR**
$FAPAR_{CAN}$ was negatively correlated with albedo in conifer dominated forests (Fig. 8). The correlation was strongest at the
smallest SZA (r = -0.91, r = -0.90) and weakest at the largest SZA (r = -0.63, r = -0.59). When including mixed plots and the
plots dominated by broadleaved trees, correlation of $FAPAR_{CAN}$ to albedo varied from almost non-existent in Alaska (r
ranging from -0.17 to 0.07) to moderate in Finland (r ranging from -0.62 to -0.30). The higher correlation in Finland can be
explained by the small number of broadleaved dominated forests in our data from Finland. In addition to the proportion of
broadleaved trees, variation in forest floor characteristics influenced the albedo-$FAPAR_{CAN}$ relations by altering the albedo
values (Fig. 8). The effect of forest floor was seen in relatively sparse canopies only. For example, at SZA of 40˚ the effect
of forest floor on albedo started to show at $FAPAR_{CAN}$ values below 0.5 (Fig. 8). Remembering that $FAPAR_{CAN}$ was tightly
related to $LAI_{eff}$, this value corresponds $LAI_{eff}$ of approx. 1. $FAPAR_{TOT}$ was strongly and negatively correlated with albedo (r
ranging from -0.97 to -0.88). The only plots that deviated from this otherwise strong relation were those Scots pine plots that
had low $FAPAR_{TOT}$ and xeric forest floor.

## 3.3 Relative importance of density and tree species

The variation in density of forests was larger in Alaska than in Finland; the 5th and 95th percentiles of basal area were 8 and
43 $m^2$ $ha^{-1}$ in Alaska, and 10 and 34 $m^2$ $ha^{-1}$ in Finland. In both study areas, decrease in basal area resulted in higher albedo
but lower $FAPAR_{CAN}$. At the smallest SZA (40°) the decrease in basal area from its 95th to 5th percentile resulted in increase
of albedo by 36% in Alaska and by 21% in Finland (Fig. 9). Correspondingly, $FAPAR_{CAN}$ decreased by 48% in Alaska and
by 44% in Finland. When SZA increased, the response of $FAPAR_{CAN}$ to basal area became weaker. For example, at SZA of
70° the basal area could be reduced to approx. 20 $m^2$ $ha^{-1}$ with equal relative changes in albedo and $FAPAR_{CAN}$ (Fig. 9b). At
the largest SZA (80°) both albedo and $FAPAR_{CAN}$ varied very little (max. 6%) between the 5th and 95th basal area
percentiles. In other words, the effect of basal area depended strongly on SZA. However, the relative decrease of $FAPAR_{CAN}$
with decreasing basal area was always larger than or equal to the relative increase in albedo.

Increasing the proportion of broadleaved trees increased the albedos considerably more than did reduction in basal area (Fig.
9c,d). The effect of broadleaved trees was slightly smaller in sparse than in dense forests. For example, at SZA of 40°,
increasing the broadleaved proportion from 0–10% to 90–100% resulted in relative increase of albedo by 130% (in Alaska)
and 80% (in Finland) in forests with high basal area (i.e., basal area percentiles from 70th to 100th). In forests with low basal
area (i.e., basal area percentiles from 0th to 30th) the corresponding figures were 112% (Alaska) and 71% (Finland). The
smaller relative increase in Finland is explained by the higher albedo of Finnish coniferous forests, because the albedos of
broadleaved species did not differ between Alaska and Finland. $FAPAR_{CAN}$ was almost independent on the proportion of
broadleaved trees, except for large SZAs where $FAPAR_{CAN}$ tended to decrease slightly when broadleaved proportion
increased (Fig. 9d). This is explained by the fact that at large SZAs $FAPAR_{CAN}$ was mainly determined by the absorption of
canopy elements, and the absorption was lower for broadleaved than for coniferous trees.

## 4 Discussion

Despite recent studies published on the relationships between albedo and boreal forest structure, and despite the widespread
use of FAPAR to monitor vegetation productivity, the physical link between forest albedo and productivity has been poorly
understood. To our knowledge, the relationship between these two quantities has not been quantified earlier for an extensive
geographical area. Another gap in the discussion has been the role of latitude: solar paths vary across the biome, and
therefore, need to be taken into account before making any generalizations on how altering forest structure through
silvicultural operations can be used to influence albedo (and furthermore, climate).

Our results show that albedo and FAPAR$_{CAN}$ are tightly linked in boreal coniferous forests. The prerequisites for this are that
there is only a limited proportion of broadleaved trees present in the forest and that the tree canopy is not very sparse (i.e.
LAI is not very low). The explanation for the tight connection between albedo and FAPAR$_{CAN}$ is that they respond with
opposite trends to forest structural variables. However, the shapes of these trends depended on directional characteristics of
the incoming solar radiation which was also reflected in the albedo vs. FAPAR$_{CAN}$ relations. This underlines the importance
of taking into account latitude and season (i.e. solar angle) when evaluating climate impacts of forests even within one
biome. FAPAR$_{TOT}$ was also tightly linked with albedo. Because FAPAR$_{TOT}$ equals one minus PAR albedo, this finding
indicates that PAR albedo and shortwave albedo of vegetation are correlated. However, the overall variation in FAPAR$_{TOT}$
was small in magnitude. Our results differ slightly from those observed by Lukeš et al. (2016) who compared satellite-based
(MODIS) albedo and FAPAR in Finland and observed much weaker (but still negative) correlation between these quantities.
The spatial resolution in their study (1×1 km) was coarser than in our study, and the FAPAR definition differed: MODIS
FAPAR is defined as PAR absorbed by green elements of vegetation canopy, both trees and understory included. In addition,
Lukeš et al. (2016) did not separate coniferous and broadleaved trees, although this effect is likely minor since the proportion
of broadleaved trees is on average low in Finland. Finally, simulation model used here, although parameterized by field
observations, cannot capture all the variability in real forests, and on the other hand, satellite products are likely to include
observation and modelling errors that increase the noise in the data.

The responses of albedo to tree species and forest structure were similar across the biome in Alaska and Finland. This
corroborates findings in previous, local studies (Amiro et al., 2006; Bright et al., 2013; Lukeš et al., 2014; Kuusinen et al.
2014; Kuusinen et al., 2016). Also the results regarding overall level of FAPAR$_{CAN}$, and the dependence of FAPAR$_{CAN}$ on
tree species were similar to earlier studies (Roujean, 1999; Steinberg et al., 2006). However, as our study was based on
extensive field data from two continents, drawing more general conclusions on how forest structure, albedo and productivity
are interconnected is now possible. In addition, to our knowledge only one study has previously evaluated the forest floor
contribution to albedo (Kuusinen et al., 2015). We showed that forest floor vegetation (which is often in practical forestry
e.g. a proxy for site fertility type) can significantly contribute to forest albedo; its average contribution can be up to 50%,
varying between forests dominated by different tree species. Similarly, the average contribution of forest floor to total
ecosystem FAPAR can be up to or even over 50%, as reported previously also by Ikawa et al. (2015) for an eddy-covariance
study site in Alaska. In other words, even though forest floor vegetation often contributes only little to, for example, total
forest biomass, it can have a significant role as a key driving factor of forest albedo and ecosystem productivity. Quantifying
the variation in forest floor composition and optical properties across the boreal biome constitutes therefore an important
research topic in the future. The important role of forest floor means also that any forest management that influences forest
floor composition can significantly alter the biophysical climate effects of forests. For example, reindeer grazing has been
suggested to reduce land surface albedo, because it reduces the cover of reindeer lichens that have higher albedo compared to
mosses (Stoy et al., 2012).

The black soil simulations that we conducted in order to quantify the contribution of forest floor explained also why the albedo increased as a function of solar zenith angle. From previous simulation studies it is known that when the sun approaches the horizon, the path length of radiation and therefore scattering from the canopy layer increase while the contribution of forest floor decreases (Kimes et al., 1987; Ni & Woodcock, 2000). The net effect is dependent on the density (gap fractions) of the canopy layer, and on the reflectance of the forest floor: if the canopy is sparse or clumped, or if the reflectance of the forest floor is high, it is likely that increasing the solar zenith angle reduces the forest floor contribution more than it increases the scattering from canopy. Our results generalize the findings of these previous studies that examined only few stands locally. It should be noted that our results apply only to summertime conditions. If the forest floor has high reflectance due to e.g. snow cover, a decrease of albedo as a function of solar zenith angle is expected to be observed more often (Ni & Woodcock, 2000).

465

We observed some interesting differences between Alaskan and Finnish datasets which deserve to be highlighted. Even though our field data do not represent a probability sample they are still well representative of the forests in the study areas. The mean albedo was higher in Alaska than in Finland, because of the higher proportion of broadleaved species in Alaska. However, the coniferous forests in Alaska had lower albedos than those in Finland. There is some previous evidence to support this, because the lowest values reported by Amiro et al. (2006) for spruce forests in Alaska are lower than those reported by Kuusinen et al. (2014) for spruce in Finland. Because the difference remained also when assuming black soil, the reason is in the properties of the canopy layer. Particularly, the low reflectance of bark in the Alaskan species (Fig. 3b) explains part of the difference.

474

Radiative transfer models offer a useful tool for assessing the radiation regime of forests, especially when the modeling approach can utilize readily available common forest inventory databases. Validating the simulated albedo and FAPAR values, however, is challenging. Even though international model intercomparison efforts such as RAMI (Widlowski et al., 2007) provide a rigorous set of reports on performance of radiative transfer models, the quality of available input data in each study where a radiative transfer model is applied is crucial. For example, the forest floor albedos that we calculated from the available reflectance spectra (Fig. 3) were clearly higher (0.18−0.23) than forest floor albedos measured in the field at other boreal sites (approx. 0.15 in Manninen & Riihelä, 2008; Manninen & Riihelä, 2009; Kuusinen et al., 2014). If we had scaled our reflectance factors in order to obtain forest floor albedos of 0.15, the simulated forest albedos would have decreased by 7–10%. Furthermore, including also the UV region in the simulations would have reduced the simulated albedos by up to 7%, assuming that the optical properties of the canopy and forest floor are similar at UV than at 400 nm. However, particularly the lack of field measured spectra for some of the Alaskan species is a limitation of our study and shows that there is an urgent need for comprehensive spectral database of boreal tree species.


Our results regarding basal area give an idea of the magnitude of the effects that varying thinning regimes could have on
forest albedo and productivity. The effect of thinnings on albedo have previously been estimated mainly by in situ
measurements at few selected sites (Kirschbaum et al., 2011; Kuusinen et al., 2014). In our study, reduction in the basal area
reduced $FAPAR_{CAN}$ equally or more compared to how albedo changed. In contrast to basal area, the proportion of
broadleaved trees had a notably larger effect on forest albedo while having only a negligible influence on forest productivity
($FAPAR_{CAN}$). The relative importance of basal area and tree species nevertheless depends on the spectral properties of the
tree species and forest floor. Based on our results, the effect of thinning (removal of basal area) on albedo and FAPAR
depends on solar angle. Therefore, the influence of thinning on forest productivity differs between latitudes. Furthermore,
because the basal area influenced albedo and $FAPAR_{CAN}$ less at large sun zenith angles, the effects of thinning integrated
over entire rotation period may not be as large as they seem when studying them only at solar noon.

Global satellite products have provided us insight on coarse-scale trends of albedo in different biomes. However, their
weakness is that even though we can establish correlations between changes in albedo and changes in land cover, we are still
not able to identify and quantify the biophysical factors which cause the albedo of a forest area to change. In addition, a
specific challenge in coupling forest management operations with changes in satellite-based albedo products is that the scale
of these operations significantly differs in North America and Northern Europe, and often does not directly correspond to the
spatial resolution of current albedo products. With an understanding of the consequences of, for example, forest management
practices on the albedo, best-practice recommendations for forest management in future climate mitigation policies will
become more justified. By coupling extensive field inventory data sets and radiative transfer modeling, we showed that
albedo and $FAPAR_{CAN}$ are tightly linked in boreal coniferous forests at stand level. However, the relation is weaker if the
forest has deciduous admixture, or if the canopies are sparse and at the same time the species composition (i.e. optical
properties) of the forest floor vary. Because the shape of the relationship between albedo and $FAPAR_{CAN}$ was shown to
depend on solar angle, studies evaluating the climate effects of forest management strategies need to consider latitudinal
effects due to varying solar paths. The comparisons between Alaska and Finland revealed that albedo and $FAPAR_{CAN}$ differ
between geographical regions because of the differences in forest structure. However, regardless of geographical region in
the boreal zone, the potential of using thinning to increase forest albedo may be limited compared to the effect of favoring
broadleaved species.
**Data availability**
Data from Co-operative Alaska Forest Inventory prior to 2009 are available at LTER Network Data Portal
(http://dx.doi.org/10.6073/pasta/d442e829a1adf7da169b6076826de563). Forest inventory data from Finland are described in
Korhonen (2011) and Majasalmi et al. (2015). Leaf and needle optical properties measured in Hyytiälä are reposited at
SPECCHIO database (http://www.specchio.ch/), and those measured in Superior National Forest are reposited at ORNL
DAAC by NASA (http://dx.doi.org/10.3334/ORNLDAAC/183). Forest floor spectra were presented in Fig. 3 of this
manuscript.

**Acknowledgments**

This study was funded in parts by the Academy of Finland projects BOREALITY and SATLASER, and by the Davis
College of Agriculture, Natural Resources & Design, West Virginia University, under the US Department of Agriculture
(USDA) McIntire–Stennis Funds WVA00106. We thank Petr Lukeš and Matti Mõttus for advice on radiative transfer
modeling, and Titta Majasalmi, Pekka Voipio, Jussi Peuhkurinen and Maria Villikka for organizing the measurements of
field plots in Finland. We also thank the School of Natural Resources and Agricultural Sciences, University of Alaska for the
establishment and maintenance of the Co-operative Alaska Forest Inventory. The forest floor reflectances at Poker Flag
Research Range were obtained under the JAMSTEC and IARC/UAF collaborative study (PI: Rikie Suzuki).

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

Table 1. Mean (standard deviation) of forest variables by dominant tree species in Alaska and Finland. The species
dominance was determined by basal area proportion: If the basal area of one of the species exceeded 80% of the total basal
area, the plot was considered to be dominated by that species. The remaining plots were labeled as mixed.

| Tree species | Number of plots | Stems per hectare | Diameter at breast height (cm) [1] | Height (m) | Crown ratio (%) [2] | Basal area ($m^2$ $ha^{-1}$) | Effective LAI ($m^2$ $m^{-2}$) [3] |
|---|---|---|---|---|---|---|---|
| Alaska | | | | | | | |
| Black spruce | 70 | 2361 (1542) | 9.3 (3.8) | 7.3 (3.2) | 69 (11) | 14.6 (9.3) | 1.0 (0.6) |
| White spruce | 124 | 806 (653) | 21.3 (7.9) | 14.7 (5.2) | 74 (9) | 22.8 (13.1) | 2.4 (1.3) |
| Quaking aspen | 22 | 1572 (916) | 15.8 (5.1) | 13.9 (3.5) | 37 (7) | 26.0 (8.8) | 2.8 (0.9) |
| Black cottonwood/ balsam poplar | 8 | 672 (658) | 35.1 (14.7) | 20.5 (5.8) | 62 (11) | 34.8 (14.5) | 2.7 (1.1) |
| Birches | 84 | 873 (662) | 22.6 (8.4) | 17.5 (2.9) | 58 (11) | 25.1 (8.1) | 3.2 (1.4) |
| Mixed | 276 | 1082 (1131) | 22.0 (8.3) | 15.1 (3.9) | 62 (12) | 25.2 (10.1) | 2.7 (1.2) |
| All | 584 | 1160 (1139) | 20.3 (9.0) | 14.4 (4.9) | 64 (13) | 23.6 (11.0) | 2.5 (1.3) |
| Finland | | | | | | | |
| Scots pine | 184 | 1165 (1301) | 18.0 (8.5) | 14.7 (6.4) | 51 (16) | 15.9 (7.7) | 1.1 (0.5) |
| Norway spruce | 115 | 980 (1014) | 19.7 (8.9) | 16.6 (6.9) | 68 (15) | 19.8 (9.4) | 2.4 (1.1) |
| Broadleaved | 23 | 1409 (1419) | 13.6 (7.1) | 13.9 (6.0) | 62 (16) | 12.6 (7.1) | 1.9 (1.2) |
| Mixed | 180 | 1094 (1782) | 20.5 (8.0) | 17.2 (5.8) | 58 (14) | 20.3 (9.1) | 2.2 (1.1) |
| All | 502 | 1109 (1444) | 19.1 (8.5) | 16.0 (6.4) | 58 (16) | 18.2 (8.9) | 1.8 (1.1) |

1) Definition of breast height differed between Alaska (1.37 m) and Finland (1.3 m).
2) Ratio of the length of living crown to tree height.
3) Not measured in the field. The values are calculated by the FRT model.
Table 2. Number of study plots by dominant tree species and forest floor type. The species dominance was determined by
basal area proportion: If the basal area of one of the species exceeded 80% of the total basal area, the plot was considered to
be dominated by that species.

| Tree species | Forest floor | | |
|---|---|---|---|
| | Grass | Shrub/moss | Lichen |
| Black spruce | 8 | 60 | 2 |
| White spruce | 13 | 111 | 0 |
| Quaking aspen | 4 | 18 | 0 |
| Black cottonwood/balsam poplar | 2 | 6 | 0 |
| Birches | 23 | 61 | 0 |
| Mixed | 40 | 236 | 0 |
| All | 90 | 492 | 2 |
| | Herb-rich | Mesic | Xeric |
| Scots pine | 2 | 145 | 37 |
| Norway spruce | 28 | 86 | 1 |
| Broadleaved | 8 | 14 | 1 |
| Mixed | 26 | 152 | 2 |
| All | 64 | 397 | 41 |


Table 3. Structural input parameters used in the FRT model simulations.

| | Leaf mass per area (g m$^{-2}$) [1] | Shoot shading coefficient [2] | Shoot length (m) [3] | Branch area to leaf area ratio [4] |
|---|---|---|---|---|
| Alaska | | | | |
| Black spruce | 187 | 0.50 | 0.05 | 0.18 |
| White spruce | 182 | 0.50 | 0.05 | 0.18 |
| Quaking aspen | 57 | 1 | 0.40 | 0.15 |
| Balsam poplar | 86 | 1 | 0.40 | 0.15 |
| Birches | 54 | 1 | 0.40 | 0.15 |
| Finland | | | | |
| Scots pine | 158 | 0.59 | 0.10 | 0.18 |
| Norway spruce | 200 | 0.64 | 0.05 | 0.18 |
| Broadleaved | 57 | 1 | 0.40 | 0.15 |

1) Black spruce and white spruce (Reich et al., 1999), quaking aspen and birches in Alaska (Bond-Lamberty et al., 2002),
balsam poplar (Sigurdsson et al., 2001), Scots pine (Palmroth & Hari, 2001), Norway spruce (Stenberg et al., 1999),
broadleaved species in Finland (values of birch from Kull & Niinemets, 1993)
2) Projected to total needle area in a shoot. Measures the effective leaf area, taking into account the self-shading of needles in
a shoot. Black spruce and white spruce (Thérézien et al., 2007), Scots pine (Smolander et al., 1994), Norway spruce
(Stenberg et al., 1995)
3, 4) Same values as used by Lukeš et al. (2013a)
Table 4. Albedo, $FAPAR_{CAN}$, and $FAPAR_{TOT}$ by dominant tree species and SZA. The reported value for given species is the
mean of plots in which the basal area proportion of that species exceeded 80%. The number of plots and mean forest
variables for each species are reported in Table 1.

| Tree species | Black-sky (SZA) | | | | | White-sky |
|---|---|---|---|---|---|---|
| | 40° | 50° | 60° | 70° | 80° | |
| Albedo | | | | | | |
| Black spruce | 0.121 | 0.122 | 0.124 | 0.128 | 0.137 | 0.124 |
| White spruce | 0.091 | 0.094 | 0.097 | 0.103 | 0.114 | 0.104 |
| Broadleaved (Alaska) | 0.194 | 0.204 | 0.218 | 0.236 | 0.262 | 0.205 |
| Scots pine | 0.144 | 0.147 | 0.152 | 0.159 | 0.172 | 0.151 |
| Norway spruce | 0.110 | 0.114 | 0.120 | 0.128 | 0.141 | 0.126 |
| Broadleaved (Finland) | 0.207 | 0.218 | 0.231 | 0.248 | 0.273 | 0.224 |
| $FAPAR_{CAN}$ | | | | | | |
| Black spruce | 0.47 | 0.53 | 0.61 | 0.72 | 0.86 | 0.53 |
| White spruce | 0.72 | 0.77 | 0.84 | 0.90 | 0.95 | 0.74 |
| Broadleaved (Alaska) | 0.78 | 0.82 | 0.86 | 0.89 | 0.91 | 0.80 |
| Scots pine | 0.50 | 0.57 | 0.65 | 0.75 | 0.86 | 0.55 |
| Norway spruce | 0.73 | 0.79 | 0.84 | 0.89 | 0.92 | 0.74 |
| Broadleaved (Finland) | 0.60 | 0.65 | 0.71 | 0.76 | 0.81 | 0.62 |
| $FAPAR_{TOT}$ | | | | | | |
| Black spruce | 0.97 | 0.97 | 0.97 | 0.97 | 0.97 | 0.97 |
| White spruce | 0.98 | 0.98 | 0.98 | 0.98 | 0.98 | 0.98 |
| Broadleaved (Alaska) | 0.95 | 0.95 | 0.94 | 0.94 | 0.93 | 0.95 |
| Scots pine | 0.97 | 0.97 | 0.97 | 0.97 | 0.96 | 0.96 |
| Norway spruce | 0.97 | 0.97 | 0.97 | 0.97 | 0.97 | 0.97 |
| Broadleaved (Finland) | 0.95 | 0.95 | 0.94 | 0.94 | 0.93 | 0.94 |


Table 5. Canopy and forest floor contributions to albedo, and forest floor contribution to FAPAR$_{TOT}$ by dominant tree
species and SZA. The reported value for given species is the mean of plots in which the basal area proportion of that species
exceeded 80%. Note that the values are directly comparable to the species specific forest albedos and FAPAR values
reported in Table 4, i.e. exactly the same plots were used to calculate the average values in both tables.

| Tree species | Black-sky (SZA) | | | | | White-sky |
|---|---|---|---|---|---|---|
| | 40° | 50° | 60° | 70° | 80° | |
| Forest albedo when assuming black soil | | | | | | |
| Black spruce | 0.053 | 0.059 | 0.069 | 0.084 | 0.108 | 0.066 |
| White spruce | 0.062 | 0.068 | 0.076 | 0.087 | 0.104 | 0.081 |
| Broadleaved (Alaska) | 0.169 | 0.182 | 0.199 | 0.221 | 0.251 | 0.186 |
| Scots pine | 0.075 | 0.084 | 0.096 | 0.114 | 0.140 | 0.094 |
| Norway spruce | 0.079 | 0.087 | 0.097 | 0.109 | 0.128 | 0.102 |
| Broadleaved (Finland) | 0.140 | 0.155 | 0.173 | 0.197 | 0.231 | 0.165 |
| Contribution of forest floor to total forest albedo, % | | | | | | |
| Black spruce | 52.9 | 48.0 | 41.4 | 32.4 | 20.2 | 46.8 |
| White spruce | 27.9 | 23.7 | 19.0 | 13.7 | 8.0 | 22.1 |
| Broadleaved (Alaska) | 12.9 | 10.9 | 8.7 | 6.5 | 4.3 | 9.3 |
| Scots pine | 45.6 | 40.6 | 34.5 | 26.8 | 17.9 | 37.7 |
| Norway spruce | 23.5 | 19.7 | 15.8 | 11.9 | 8.0 | 19.0 |
| Broadleaved (Finland) | 32.7 | 29.5 | 25.9 | 21.9 | 17.1 | 26.3 |
| Contribution of forest floor to FAPAR$_{TOT}$, % | | | | | | |
| Black spruce | 50.1 | 44.1 | 36.0 | 25.1 | 11.1 | 45.7 |
| White spruce | 26.4 | 20.6 | 14.5 | 8.3 | 2.6 | 24.3 |
| Broadleaved (Alaska) | 16.9 | 12.5 | 8.3 | 4.6 | 2.0 | 15.9 |
| Scots pine | 46.3 | 39.8 | 31.7 | 21.5 | 10.5 | 42.8 |
| Norway spruce | 24.4 | 18.7 | 13.2 | 8.3 | 4.4 | 23.3 |
| Broadleaved (Finland) | 34.7 | 29.3 | 23.5 | 17.7 | 12.4 | 34.3 |


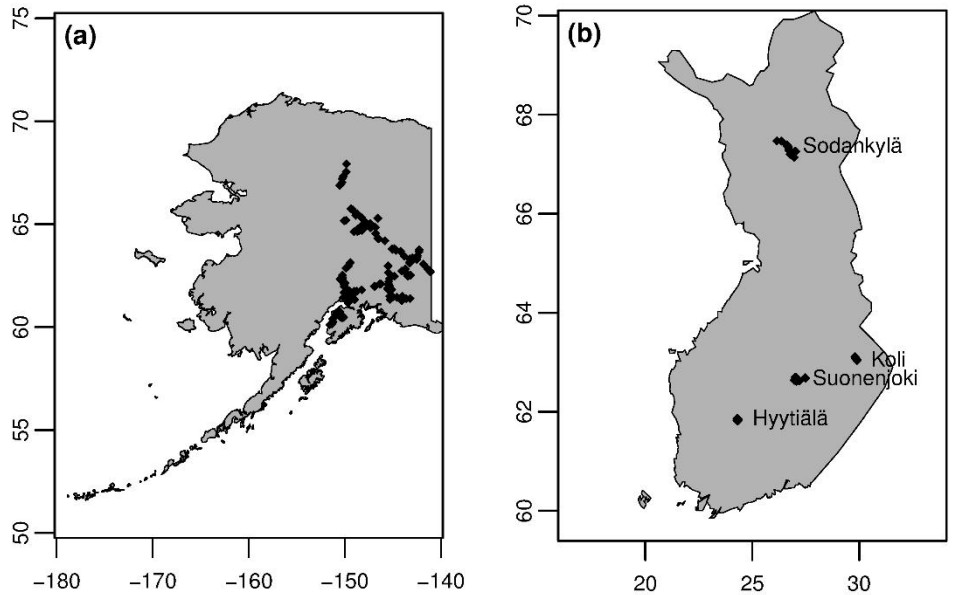

Figure 1. Location of the field plots.

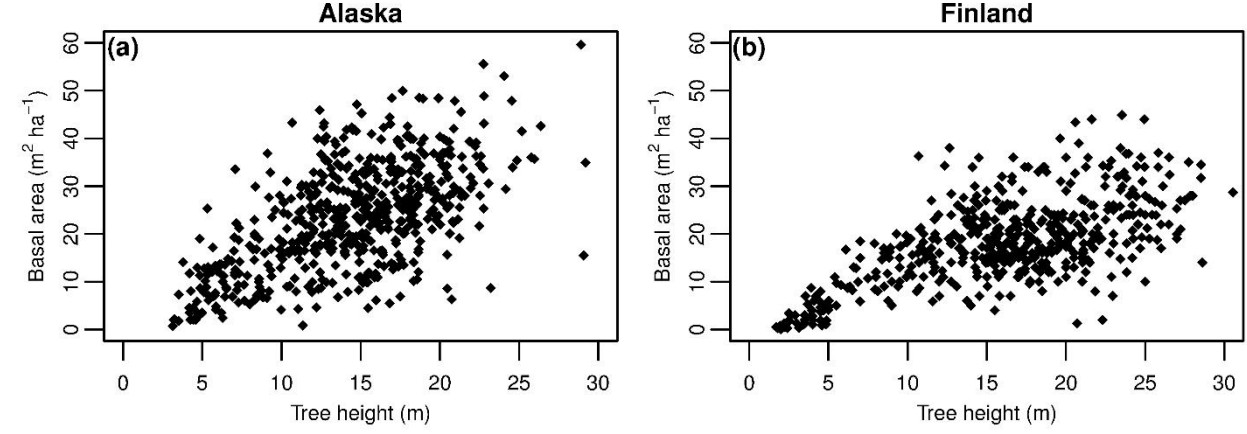


Figure 2. Basal area against tree height in the study plots in Alaska (a) and Finland (b).

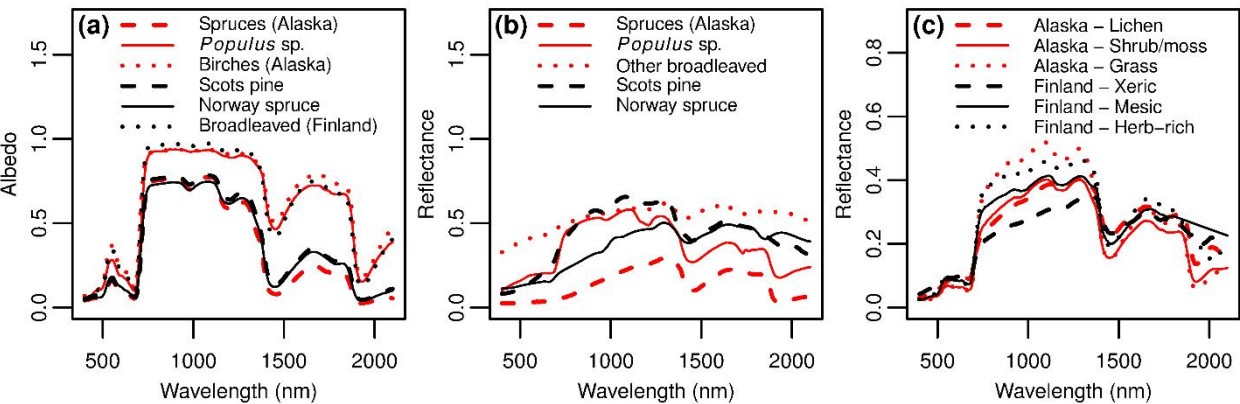


Figure 3. Spectra of vegetation elements used in the simulations: (a) leaves/shoots, (b) bark, (c) forest floor. The values for
leaf and shoot are single scattering albedos (reflectance + transmittance), and the values for bark and forest floor are
reflectance factors.

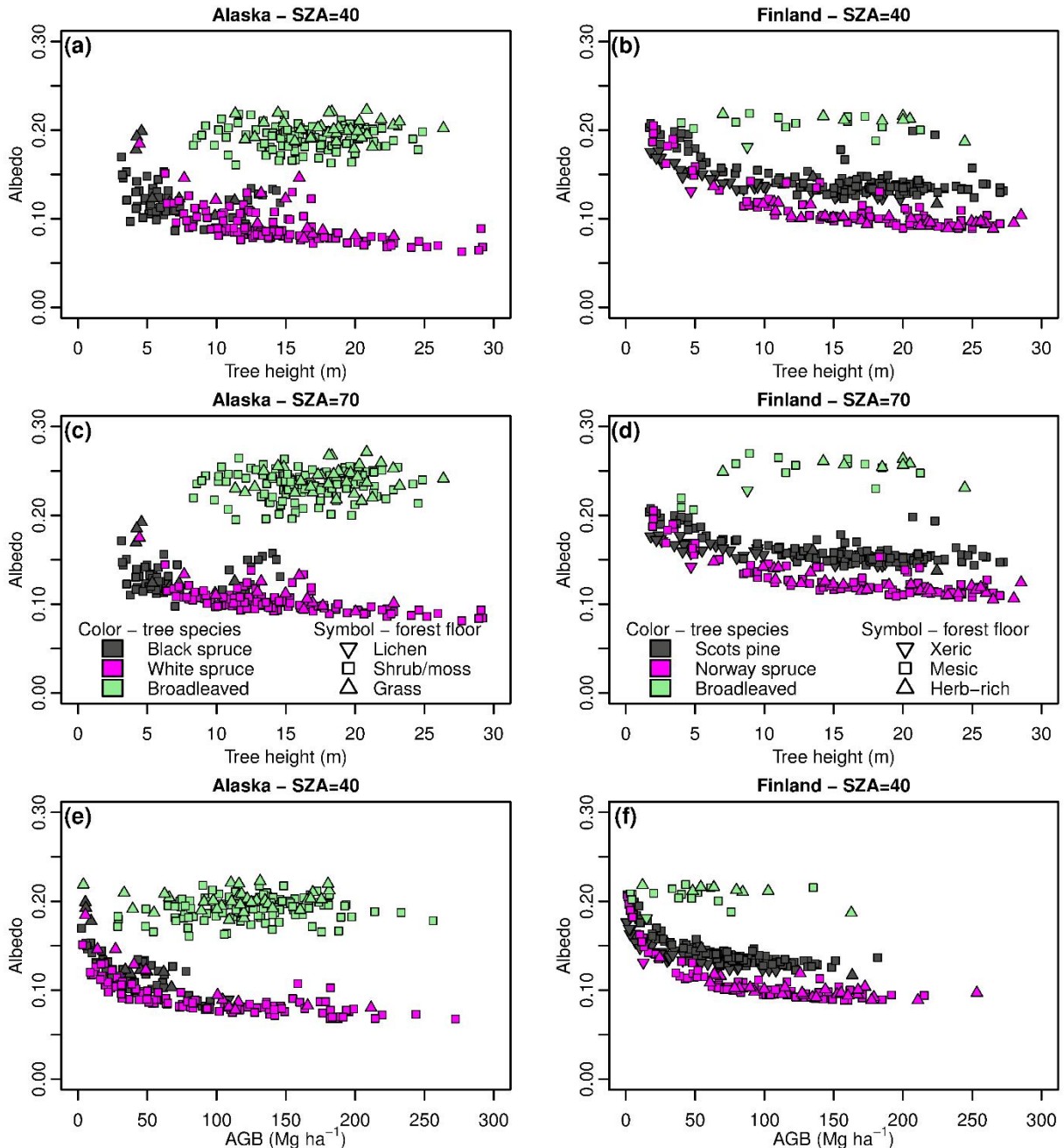

Figure 4. Forest black-sky albedo as a function of tree height (a–d) and AGB (e–f). Relations to tree height are shown for two SZAs, 40° (a–b) and 70° (c–d), representing solar noon at midsummer and the annual average in the study regions. Left hand column shows the results for the Alaskan data, and right hand column for the Finnish data. The figures show only monospecific plots, i.e. plots in which the basal area proportion of one of the species exceeded 80%.

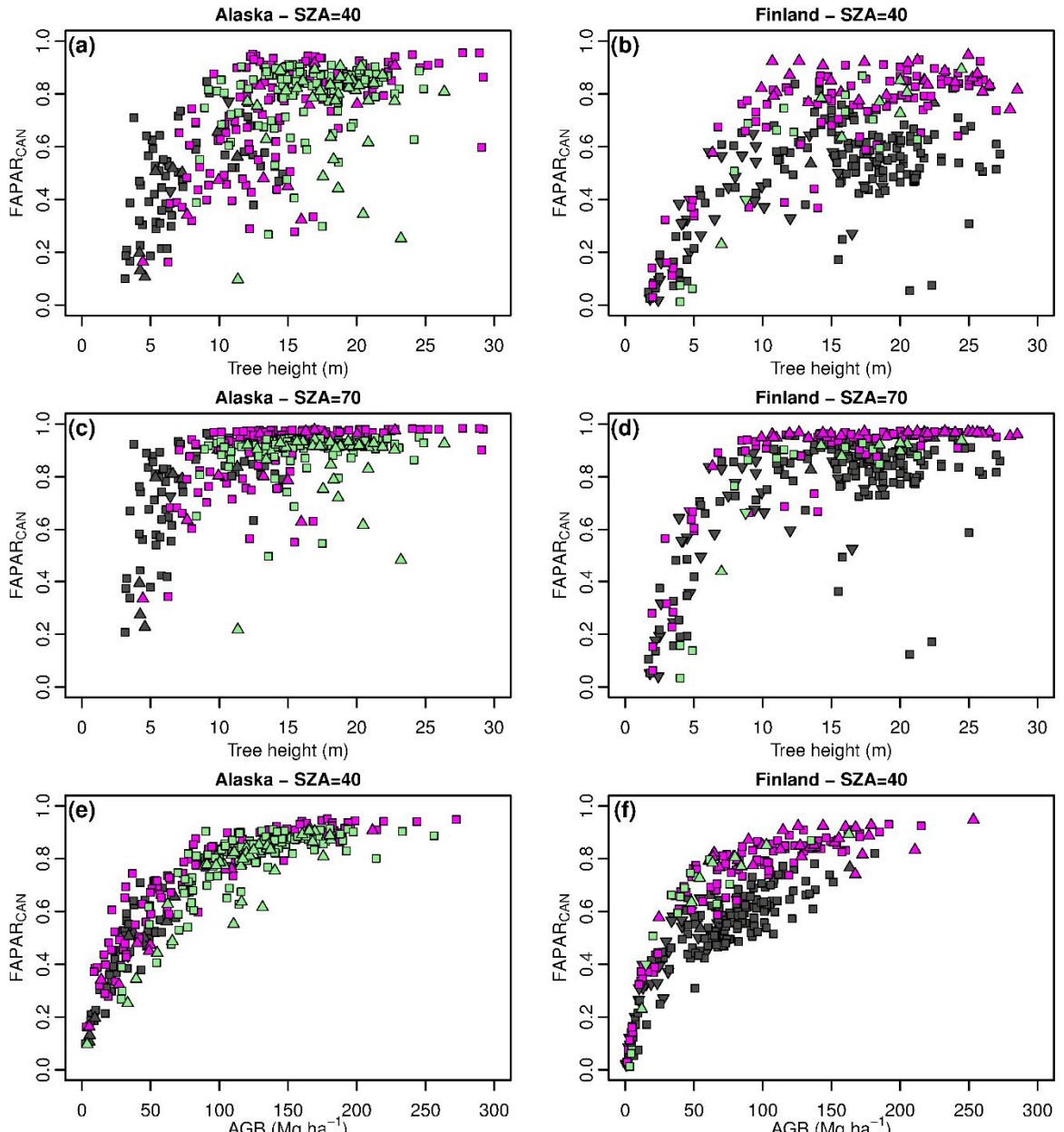


Figure 5. Black-sky FAPAR$_{CAN}$ as a function of tree height (a–d) and AGB (e–f). Relations to tree height are shown for two
SZAs, 40° (a–b) and 70° (c–d), representing solar noon at midsummer and the annual average in the study regions. Left hand
column shows the results for the Alaskan data, and right hand column for the Finnish data. The figures show only
monospecific plots i.e. plots in which the basal area proportion of one of the species exceeded 80%. For explanation of the
symbols, see legend in Fig. 4.

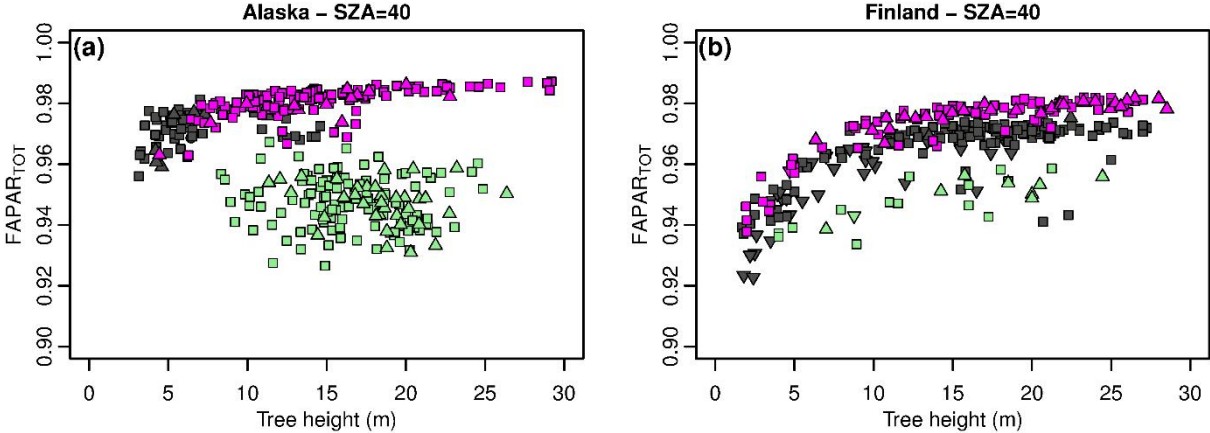


Figure 6. FAPAR$_{TOT}$ as a function of tree height at SZA of 40°. The figures show only monospecific plots i.e. plots in which
the basal area proportion of one of the species exceeded 80%. For explanation of the symbols, see legend in Fig. 4.

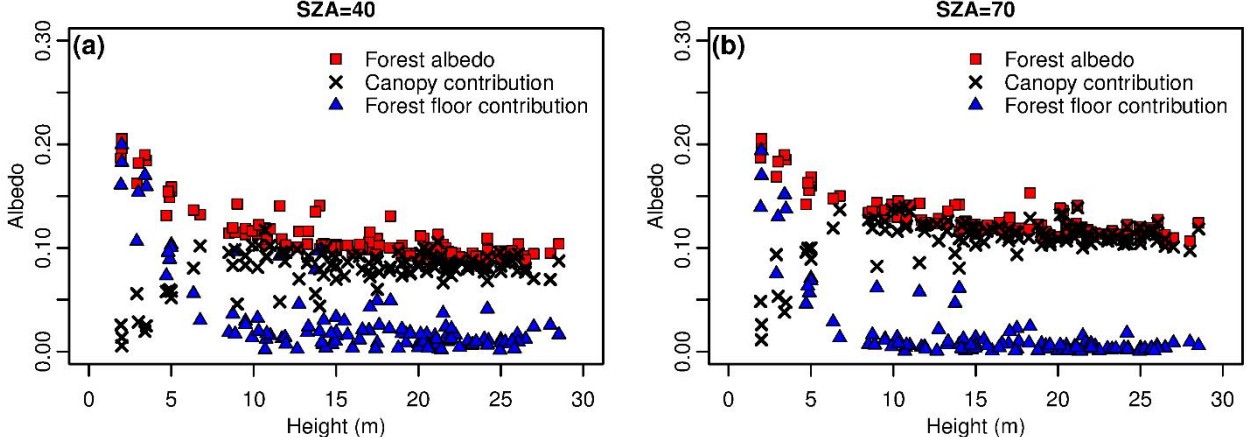


Figure 7. Canopy and forest floor contributions to forest black-sky albedo as function of tree height. Canopy contribution
was obtained by assuming black soil in the simulation. Forest floor contribution was obtained by subtracting the canopy
contribution from the total forest albedo. The data shown are from Norway spruce dominated forests in Finland.

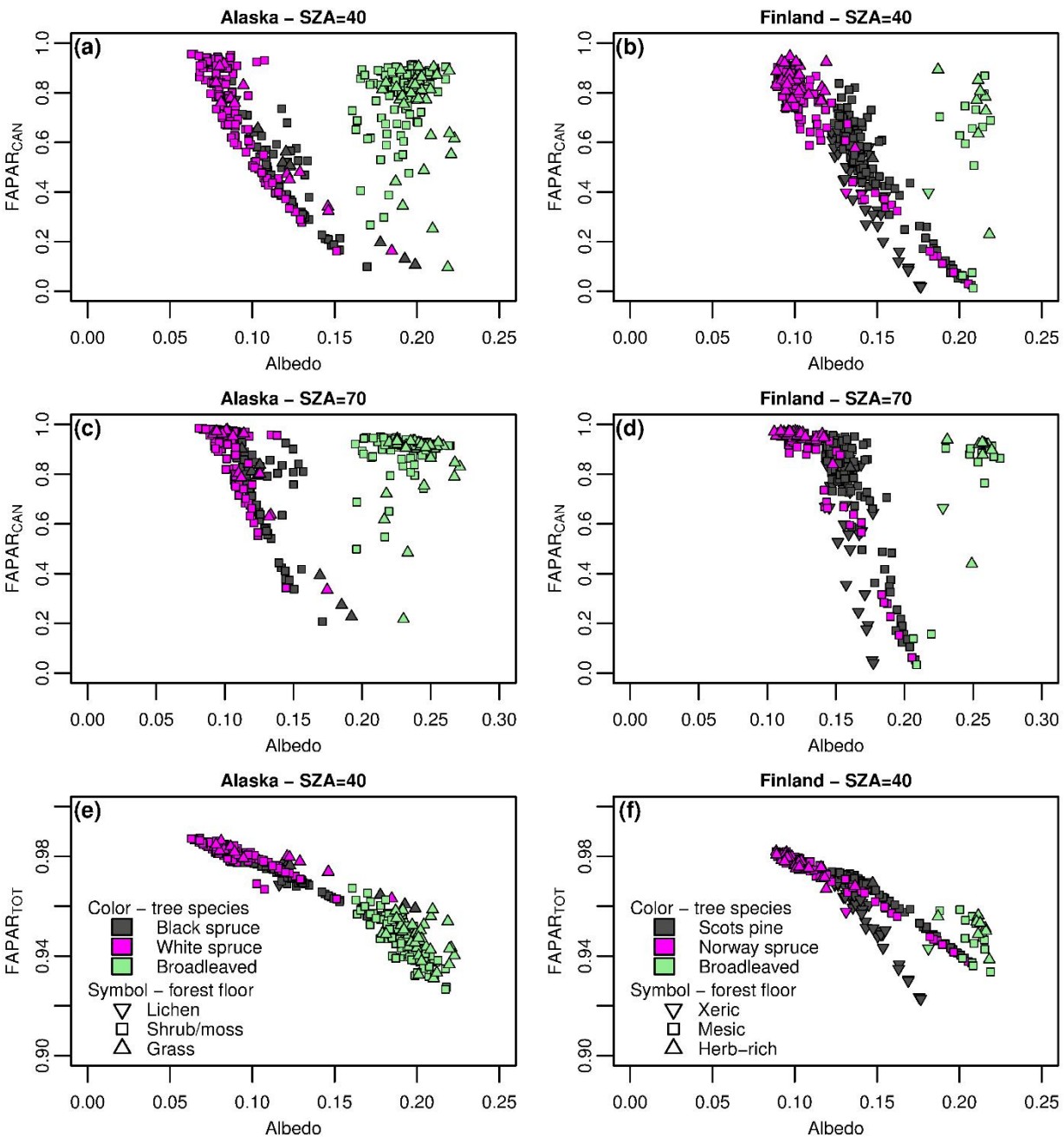

Figure 8. Relation of FAPAR to forest black-sky albedo by dominant tree species. The figures show only plots that were dominated by one species i.e. in which the basal area proportion of one of the species exceeded 80%. a–d: $FAPAR_{CAN}$ against albedo at two SZAs, 40° and 70°, representing solar noon at midsummer and the annual average in the study regions; e–f: $FAPAR_{TOT}$ against albedo at SZA of 40°.

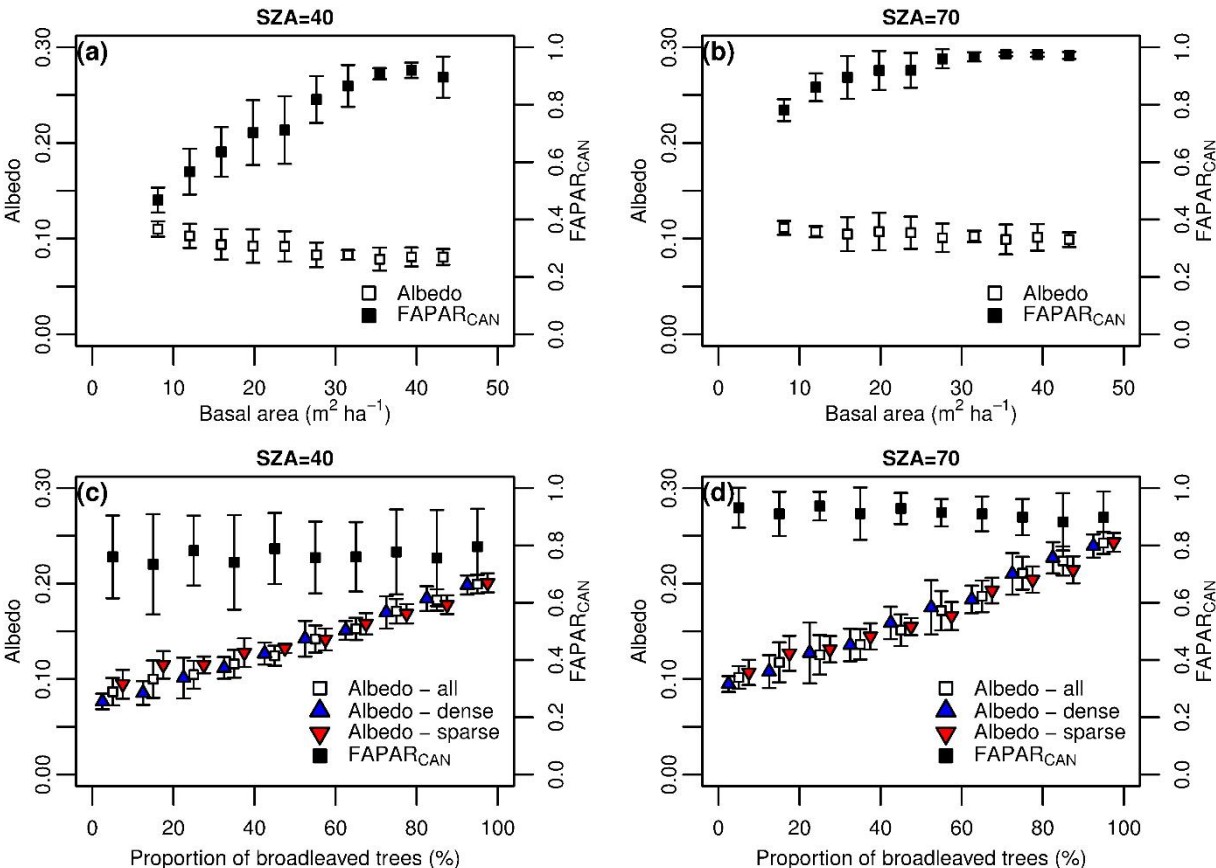


Figure 9. Effect of basal area (a–b) and proportion of broadleaved trees (c–d) on black-sky albedo and FAPAR$_{CAN}$ at sun
zenith angles of 40° and 70° in Alaska. Points represent mean and whiskers the standard deviation in ten equally spaced
classes. Effect of broadleaved proportion on albedo is presented separately for dense (basal area > 31 m$^2$ ha$^{-1}$) and sparse
(basal area < 21 m$^2$ ha$^{-1}$) forest. These limits correspond to 30th and 70th percentiles of basal area in Alaskan data. The
points representing dense and sparse forest are shifted along the x axis in order to make them visible.