# Peer review of "Quantifying the missing link between forest albedo and productivity in the boreal zone"

_Biogeosciences, 2016_

## Referee Comment (RC1) · Anonymous Referee #1 · 14 Jun 2016

This paper presents an assessment of the impact of forest structure (type of tree & broadleaf v deciduous proportion) on albedo and hence FAPAR as a proxy for productivity. This is an important topic given the link between productivity and climate and the use of remote sensing to estimate albedo across large areas. The paper is very well written, clear and the results are well presented. I have a few queries regarding the methods, particularly sensitivity and generality, but if the authors can address these then the paper is suitable to publish and would be of wide interest.

One general query is the model sensitivity to choice of structural assumptions and parameters. It's not clear to me that there is any real effort made to quantify the sensitivity of the results to the assumptions of crown shape, and crown leaf area density. Tree crowns vary a lot in shape, are heavily clumped, and leaf size, angle and woody material have a big impact on the BRF. It would be good if the authors could quantify the

impacts of some or all of these assumptions on the results. They use tree classes but how big is within and between class variability? The issue is the FRT parameters are driven by allometrics, but these are likely to be very specific aren't they? Hence my comments about generality below.

Similarly, the authors show the importance of the understory, particularly with view and sun angle. Can they say more about this given that in many areas understory can be very significant and can be correlated in terms of cover with the overstory?

The authors are making a claim for generality based on the number of plots they have and the ranges of cover and density and deciduous v conifer mix they have. However I would question in particular how general the Finnish birch forests are likely to be - how representative of deciduous broadleaf forests? Can the authors justify this aspect better?

---

## Author Comment (AC1) · 20 Jun 2016

Due to technical reasons, Figure 6 appears two times in the manuscript (as Figure 6 and then again as Figure 7). The figure that was intended to be nr 7 is missing entirely. Please find the correct version of Figure 7 attached. The figure captions are correct.

On behalf of co-authors, Aarne Hovi

[Figure]

**Fig. 1.**

---

## Referee Comment (RC2) · Anonymous Referee #2 · 6 Jul 2016

The study by Hovi et al. is addressing the important topic of how forest management and composition is influencing albedo and fapar. The understanding and quantification of the relation of albedo and fapar are prerequisites for assessing the effectiveness of forest management for climate mitigation, while including the radiative forcing effect through the energy budget. The study complements observational studies through radiative transfer modelling. Results reveal that radiative forcing can be reduced through increased albedo by increasing the abundance of deciduous species. The study is an important contribution towards a better understanding of forest structure on albedo and FAPAR, thus linking two main components of the climate, i.e. the energy and carbon cycle.

While the topic is interesting and important, the study has major shortcomings.

1. The study is based on bidirectional radiation quantities for albedo (black sky albedo), no diffuse irradiance is taken into account. At the high latitudes of the test sites, the fraction of diffuse radiation cannot be neglected. The effect of varying leaf angles might significantly decrease under a scenario with diffuse irradiance. I expect that the results (difference between broadleaf and needleleaf) might be much less significant when introducing a realistic diffuse fraction. If the study is supposed to serve as a baseline for future management, it needs to quantify differences under realistic irradiance scenarios for the given latitudes.

2. The study assumes that fapar is a proxy for productivity. This assumption (and related study title) is too simplistic as light is only one of several growth limiting factors, and light use efficiency needs to be accounted for at the species or plant functional type level. Also other limiting factors such as temperature, soil water, and vapor pressure deficit would need to be accounted for at the species or plant functional type level for the conversion of fapar to GPP. Further, productivity in sunlit and shaded leaves is not linearly scaling with APAR (see light saturation curve).

3. The definition and usage of fapar is unclear – when using fapar for GPP estimation, only fapar absorbed by leaves is relevant. Forest canopy fapar is not mainly determined by leaf area index and directionality of incoming solar radiation (as stated in line 64), but – depending on the fraction of leaf to plant area, very much by stems, branches, and the understory. It is mentioned that no correction was done for litter, but it is unclear if the same is true for stems, branches, and understory (which might contain open soils, lichen, etc.).

4. Equations section of albedo and fapar – both quantities are not fluxes (of radiation), but ratios! Review definitions and revise equations. Also, explain how spectral weighting based on TOA spectral distribution is influencing results compared to weighting by top of canopy irradiance spectral distribution.

---

## Author Response (AR1)

Dear Dr. Luyssaert,

We have revised our manuscript. Please find below first our response to your concerns, followed by responses to referees #1 and #2, and finally list of changes and the revised manuscript with tracked changes. Our responses are indented and in blue font.

On behalf of co-authors,
Aarne Hovi

Associate Editor Decision: Reconsider after major revisions (24 Aug 2016) by dr. Sebastiaan Luyssaert

Comments to the Author:
Dear authors,

Thank you for your contribution to the on-line discussion in which you have addressed most of the reviewer's comments. Based on your discussion and my own reading of the paper, I would like to invite you to prepare a revised manuscript. While revising the manuscript, please, consider the following comments in addition to the concerns raised by the reviews:

- Reviewer 1 raised an important issue on the sensitivity of the model. Although I find the content of your response satisfactory, I would like to see this issue addressed by means of a comprehensive discussion. For example, a table showing for which parameters the sensitivity has already been tested (and the sensitivity itself) and for which parameters the sensitivity has not yet been tested.

> We have expanded our response to reviewer #1's comment. As in our first response, we refer to the studies that have performed sensitivity analyses, but we now discuss more thoroughly how the model results were affected in those studies.

> Regarding the parameters for which sensitivity analyses have not been published, we discuss the expected sensitivity from a theoretical point of view, and based on a small sensitivity analysis that we performed for three example stands (pine, spruce, and birch) in Finland (see Appendix 1 of this document).

- Please, discuss the validity of the model to simulate absorbed radiation (see above). Is the absorbed fraction calculated as the residual of the reflected and the transmitted fraction? Has absorbed light been validated against field measurements?

> Absorbed fraction is calculated from the incoming and outgoing fluxes on-top-of and below canopy. We describe this in Section 2.2.1 of the manuscript. To our knowledge absorbed fraction has not been validated against field measurements, but transmitted and reflected fractions have been compared against other models in RAMI exercise (Widlowski et al. 2007), in which FRT has been among the best performing non-Monte Carlo models.

- Reviewer 2 raised the issue of understory and the possible role of lichens in the boreal zone. There is quite some literature on the effect of lichens on albedo. Please, discuss this issue in the light of forest management (see for example studies linking reindeer grazing - which is a non-wood forest use - to lichens, soil freezing and albedo).

> We added a note about the role of forest management effects on forest floor composition in the Discussion Section where we discuss the role of understory in the forest albedo and FAPAR. We added a reference to Stoy et al. (2012) who reported the effect of reindeer grazing on lichens and albedo.

Anonymous Referee #1

This paper presents an assessment of the impact of forest structure (type of tree &
broadleaf v deciduous proportion) on albedo and hence FAPAR as a proxy for productivity.
This is an important topic given the link between productivity and climate and the
use of remote sensing to estimate albedo across large areas. The paper is very well
written, clear and the results are well presented. I have a few queries regarding the
methods, particularly sensitivity and generality, but if the authors can address these
then the paper is suitable to publish and would be of wide interest.

One general query is the model sensitivity to choice of structural assumptions and parameters.
It's not clear to me that there is any real effort made to quantify the sensitivity
of the results to the assumptions of crown shape, and crown leaf area density. Tree
crowns vary a lot in shape, are heavily clumped, and leaf size, angle and woody material
have a big impact on the BRF. It would be good if the authors could quantify the impacts of some or all of these
assumptions on the results. They use tree classes but how big is within and between class variability? The issue
is the FRT parameters are driven by allometrics, but these are likely to be very specific aren't they?

We acknowledge that the choice of parameters has an influence on the exact values of the simulated
quantities in any modeling study. The main aim of our study is to describe the general relations
between forest structure, albedo, and FAPAR. Such studies have been limited by the low resolution of
remote sensing data and, on the other hand, by the lack of extensive in situ measurement data.
Therefore the modeling approach is a good (and often the only) option for studying albedo and FAPAR
in a large variety of forests, yet maintaining high spatial resolution. The advantage of FRT, when
compared to many other theoretical radiative transfer models, is that it can be parameterized with
standard forest inventory data and allometric models. These were used also in the current study,
because the forest field inventories did not directly measure the required parameters for radiative
transfer simulation (e.g. LAI, canopy cover).

FRT is one of the most well-known forest radiative transfer models and was originally published already
in 1991 (Nilson and Peterson, 1991) and later modified by Kuusk and Nilson (2000). There are already a
number of studies reporting the sensitivity of the FRT model to its input parameters. FRT has also
participated in the large international "Radiation transfer model intercomparison" (RAMI) effort
coordinated by JRC, and thus its performance in relation to other models has been documented in a
wide range of tests. The results from these tests are publicly available online (http://rami-
benchmark.jrc.ec.europa.eu/HTML/) and reported in peer-reviewed scientific papers by Widlowski et al.
Therefore, we have not reported sensitivity analyses in this manuscript.

The reviewer asked about the assumptions related to crown shape and their influence on the results.
The effect of crown shape on forest BRDF characteristics was quantified in Rautiainen et al. (2004). In
the studied forest stands, assuming conical crowns resulted the forest hemispherical-directional reflectance to be 62%-98% (red), 77%-99% (near infrared), and 79%-98% (mid infrared) of the value obtained when assuming ellipsoid crown. The volume of conical crown was half of the volume of ellipsoid one. They also noted, that reducing the crown volume reduces scattering from crowns but increases scattering from the forest floor. The net effect is therefore either negative or positive, depending on whether scattering from crowns or scattering from forest floor dominates the total forest reflectance. In other words, the effect of crown shape depends on canopy closure. Based on Rautiainen et al. (2008) cone is least accurate for estimating crown volume of Scots pine and Norway spruce, whereas the differences between other crown shapes were minor.

The performance of a wide range of foliage mass and crown radius (i.e. canopy/crown closure) models in forming the input of FRT was reported by Lang et al. (2007), by comparing the simulated forest reflectance factors to actual satellite observations. Regarding foliage mass models, the conclusion was that simple foliage mass models that are based on tree stem diameter at breast height, predicted larger variability and resulted in slightly higher correlation between the simulated and measured reflectance factors than the multiple regression models that included tree height and crown depth. On the other hand, the authors noted that the estimates from the simple regression models can be biased. Regarding crown radius, models that account for stand density effects performed better compared to the regression equations based on the breast height diameter only. In reality one has to consider also the availability of the models for the study area. The simple linear models based on stem diameter, that we used for Finnish plots, were based on data that was geographically closest to our study area in Finland. For the Alaskan species, there were also versions that took into account the stand density, but their predictive power, measured as explained variability ($R^2$) in the crown radius, was only slightly better than the simpler models.

For tree size distribution no sensitivity analyses have been published. In theory, a forest with trees of very different sizes would have a higher canopy surface roughness, which would in turn lead to somewhat lower reflectance (albedo) values. The reason for not including several tree size classes was that there were no field measurements made on tree size distribution in the data from Finland. Stand basal area was measured in the field, but the stem diameter and tree height were measured only for a basal area median tree. For Alaskan data each tree was measured individually, and thus information on the size distribution was available. It was however not taken into account because we wanted to maintain the same calculation procedure for both study areas, in order not to introduce any differences due to data processing steps.

We added references to the publications mentioned above in Section 2.2.2.

Hence my comments about generality below.

Similarly, the authors show the importance of the understory, particularly with view and sun angle. Can they say more about this given that in many areas understory can be very significant and can be correlated in terms of cover with the overstory?

Yes, the density and species richness of forest floor vegetation may vary a lot, and may be correlated
with the density of overstory layer. To be able to assess the contribution of forest floor vegetation to
forest albedo in more detail, quantitative data on forest floor composition and spectral data on all of
the components would be needed. Field data on forest floor species composition was available only for
some of our study plots, but not for all of them. In our study sites where the data was available, the
correlation between overstory and forest floor vegetation coverage was rather weak (Alaska r = -0.27;
Hyytiälä (Finland) r = -0.33). More importantly, we did not have optical properties measurements for
each of the forest floor components (litter, bare soil, various plant species) separately. Therefore, we
decided not to model the forest floor in more detail than it was done in the current version.

Overall, the large contribution of understory implicates that, in addition to tree canopy structure, the
species composition of understory has an important role in controlling boreal forest albedo and FAPAR.
Therefore, future studies should aim at more accurate characterization of forest floor composition and
optical properties. We added a note about this in the discussion where we discuss the importance of
forest floor on albedo and FAPAR.

The authors are making a claim for generality based on the number of plots they have
and the ranges of cover and density and deciduous v conifer mix they have. However
I would question in particular how general the Finnish birch forests are likely to be -
how representative of deciduous broadleaf forests? Can the authors justify this aspect
better?

It is true that our study is limited by the available field data. One of the main ideas is to compare
intensively managed (Finnish) with more naturally grown (Alaskan) forests. We have noted in our
discussion that the data is not a probability sample, and we do not claim that the results would be
applicable as such to entire boreal zone.

Regarding the broadleaved species; they existed in both Alaskan and Finnish forests, and the results
were similar in both regions. Therefore we consider that the broadleaved species in the boreal zone are
quite well represented.

We checked the wordings throughout the text and modified them when necessary to avoid making the
impression that too broad generalizations were made.

Anonymous Referee #2

The study by Hovi et al. is addressing the important topic of how forest management
and composition is influencing albedo and fapar. The understanding and quantification
of the relation of albedo and fapar are prerequisites for assessing the effectiveness
of forest management for climate mitigation, while including the radiative forcing effect
through the energy budget. The study complements observational studies through radiative
transfer modelling. Results reveal that radiative forcing can be reduced through
increased albedo by increasing the abundance of deciduous species. The study is an
important contribution towards a better understanding of forest structure on albedo and
FAPAR, thus linking two main components of the climate, i.e. the energy and carbon
cycle.

While the topic is interesting and important, the study has major shortcomings.
1. The study is based on bidirectional radiation quantities for albedo (black sky albedo),
no diffuse irradiance is taken into account. At the high latitudes of the test sites, the
fraction of diffuse radiation cannot be neglected. The effect of varying leaf angles might
significantly decrease under a scenario with diffuse irradiance. I expect that the results
(difference between broadleaf and needleleaf) might be much less significant when
introducing a realistic diffuse fraction. If the study is supposed to serve as a baseline for
future management, it needs to quantify differences under realistic irradiance scenarios
for the given latitudes.

In the first version of the manuscript, we chose to simulate direct illumination only i.e. black-sky
conditions, because this way the simulated albedo and FAPAR are independent of the parameterization
of the atmosphere. Introducing the effect of atmosphere would make the analysis more complicated,
and any differences in the modeled atmosphere, whether real or caused by uncertainty in the chosen
parameters values, would affect the comparisons between the study regions located on two continents.
We wanted to avoid this because the focus was on modeling the effects of forest structure.

However, we acknowledge that the effect of angular distribution of incoming solar radiation is
important, as seen already in the differences between sun zenith angles when assuming black-sky
conditions. In general, presence of atmosphere (and clouds) would reduce the relative share of short
(blue) wavelengths in the incoming solar spectrum due to scattering and ozone absorption. On the
other hand clouds would also increase the absorption by water vapor, which occurs in longer
wavelengths. Because reflectivity of vegetation is higher in the infrared than in the visible region, these
two phenomena have opposite effects on the simulated forest albedo and therefore would probably
cancel out each other.

Therefore, as suggested by the reviewer, we repeated the simulations in white-sky conditions i.e.
assuming totally isotropic incoming radiation. However, we retained the top-of-atmosphere solar irradiance spectrum, because modeling the effect of clouds would be highly dependent on prevailing
cloud conditions (e.g., thickness and altitude of clouds), about which we did not have measurement
data.
Simulated white-sky albedo was similar to black-sky albedo at sun zenith angles of 50°–70°, which is
logical because in the case of isotropic illumination the albedo is weighted average of the black-sky
albedos at all possible SZAs. The dominant tree species influenced which sun-zenith angle in the black-
sky case best corresponded to the white-sky albedo. $FAPAR_{CAN}$ in white-sky conditions was similar to
$FAPAR_{CAN}$ in black-sky conditions at SZAs of 40°–50°. Similarly as for albedos, tree species influenced
whether the black-sky $FAPAR_{CAN}$ at SZA of 40° or the one at 50° was closer to the white-sky case. In
general, the differences between tree species in the white-sky case did not drastically change compared
to black-sky cases. Rather than leaf angles, the species differences are mainly caused by the differences
in leaf area index (visibility of forest floor) and in the spectral properties of the canopy elements. These
factors have an influence at all angles of illumination, although some angular dependences may exist.
In the revised version, we report the simulated white-sky albedos (species-specific mean white-sky
albedos and FAPARs were added in Tables 4 and 5). We also added to Section 2.2.1 explanations of how
the white-sky albedos and FAPARs were calculated.
2. The study assumes that fapar is a proxy for productivity. This assumption (and
related study title) is too simplistic as light is only one of several growth limiting factors,
and light use efficiency needs to be accounted for at the species or plant functional type
level. Also other limiting factors such as temperature, soil water, and vapor pressure
deficit would need to be accounted for at the species or plant functional type level for
the conversion of fapar to GPP. Further, productivity in sunlit and shaded leaves is not
linearly scaling with APAR (see light saturation curve).
We agree that all these factors (and many others including e.g. diffuse to total irradiance ratio) affect
gross primary productivity, which can be estimated from FAPAR and light use efficiency (LUE), using the
well-known model by Monteith et al. (1972). Usually FAPAR can be estimated from remote sensing data
since it is directly linked to the radiation reflected by the vegetation canopies. LUE, on the other hand,
varies dynamically over different time scales, and depends on the physiological condition of the
vegetation. It is true that LUE can be different for different plant functional types and e.g. for different
tree species. However, to be able to take LUE into account in the analysis would require to model
realistically its dependence on all the mentioned environmental factors, which was not possible and
would have added a major uncertainty component into the analysis. Our study relies on the fact that
although modified by the efficiency by which plants use the absorbed PAR radiation, ultimately the
photosynthesis is driven by the absorbed PAR. Furthermore, rather than giving the exact numbers of
GPP, we see that the main value of our study is that it adds basic understanding on how albedo and the
solar energy used for photosynthesis are connected in differently structured forests. Linking GPP with
albedo would require completely different approach, probably utilizing field measurements of $CO_2$
fluxes or tree growth directly. Alternatively, statistical growth and yield models could be used.

Nevertheless, we added to the introduction a note that in addition to FAPAR, productivity is affected
also by LUE.

3. The definition and usage of fapar is unclear – when using fapar for GPP estimation,
only fapar absorbed by leaves is relevant. Forest canopy fapar is not mainly determined
by leaf area index and directionality of incoming solar radiation (as stated in line 64), but
– depending on the fraction of leaf to plant area, very much by stems, branches, and
the understory. It is mentioned that no correction was done for litter, but it is unclear if
the same is true for stems, branches, and understory (which might contain open soils,
lichen, etc.).

All quantities reported in the manuscript are total, including both green and woody/dead biomass
components. We agree that green FAPAR would be more justified in terms of productivity, but it was
not possible to separate here, because no measurements on fraction of branch area to leaf area were
made in the study plots. The same applies to the cover of litter on the forest floor which was available
for some of the field plots but not for all of them. Concerning the forest floor, we would also like to note
that open soils are very rarely seen in boreal forests where the floor is covered by (at least) green
mosses.

We added to Section 2.2.1 a note that similarly as for FAPAR$_{TOT}$, also in the case of FAPAR$_{CAN}$ green
biomass was not separated.

4. Equations section of albedo and fapar – both quantities are not fluxes (of radiation),
but ratios! Review definitions and revise equations. Also, explain how spectral weighting
based on TOA spectral distribution is influencing results compared to weighting by
top of canopy irradiance spectral distribution.

In our study the fluxes are equal to ratios, because the incoming radiation in FRT simulation equals one.
We changed the notation in the equations and in the text in Section 2.2.1. We now use terms "upward
scattered fraction of incoming radiation" ( $f_\lambda$ ⁻ ) and "downwelling (directly transmitted or downward
scattered) fraction of incoming radiation" ( $f_\lambda$ ⁻ ).

The explanation for using top-of-atmosphere spectrum is the same as in our response to Question #1
above, i.e. we wanted to avoid the uncertainty in modeling the atmosphere. However, to demonstrate
the effect of using at-ground solar spectrum, as requested by the reviewer, we performed simulations
also in blue-sky conditions, modeling the effect of atmosphere on the incoming solar irradiance (direct
and diffuse components) using the SPCTRAL2 model by Bird & Riordan (1986). The model assumes clear
skies (no clouds), and requires as parameters ozone and water vapor concentration, as well as aerosol
optical depth (AOD). We used ozone and water vapor from the U.S. standard atmosphere, and AOD
from measurement data. The values are the same as used for ASTM standard solar spectrum (ASTM
standard G173-03). Because the data do not represent the study areas, the results are intended for
demonstration purposes only. We repeated the blue-sky simulations at all SZAs (40° to 80°).

For all SZAs, the blue-sky albedo was very highly correlated (r = 1.00) with black-sky albedo, as expected. At small SZAs (40° to 60°), also the overall level of blue-sky albedo was almost equal to that of black-sky albedo (mean difference of up to 0.006 in absolute units). At SZAs of 70° and 80° the blue-sky albedos were somewhat higher than black-sky albedos (mean differences of 0.012 and 0.031 in absolute units). This was because at high SZAs the irradiance distribution of solar spectrum was shifted towards longer wavelengths in which the vegetation is more reflective. However, due to high correlation with black-sky albedo, the conclusions regarding species differences and the effect of forest structure on albedo would remain the same even if assuming blue-sky conditions. The conclusions regarding FAPAR were the same as those regarding albedo, i.e. assuming blue-sky would slightly affect the overall level of FAPAR at large SZAs while at low SZAs the level of FAPAR would remain almost the same.

We added a short summary of the results described above to end of Section 2.2.1, and denoted that because of the high correlation between black-sky and blue-sky results, the use of at-ground solar spectrum would not change our conclusions. All results reported in Section 3 are based on top-of-atmosphere solar spectrum.

Appendix 1. Theoretical discussion and sensitivity analysis on the effect of FRT parameters.
Below we discuss how the FRT parameters would affect simulation results. In addition, sensitivity of FRT
on its input parameters was tested in three example stands in Finland. The stands were selected so that
the forest variables are close to average species-specific forest variables in our data from Finland.
Tree distribution parameter would influence the visibility of the forest floor, more clumped distribution
patterns rendering larger portion of the forest floor visible. On the other hand, the scattering from tree
canopy would be larger when the trees are more regularly distributed. In our sensitivity analysis the
albedo decreased by 4–18% when the tree distribution parameter increased from 0.9 to 1.4. The effect
was largest in the spruce stand that had the largest LAI.
Leaf mass per area (LMA) would directly affect the leaf area index values. In our sensitivity analysis the
albedo was reduced by 0–9% when the LMA increased from 70% to 130% of its default value.
Shoot shading coefficient would affect the results in exactly the same manner as the leaf mass per area,
since it influences the effective LAI. It should be noted that the literature values of shoot shading
coefficients for conifers varied between 0.50–0.65, and therefore the effect of shoot shading coefficient
would be small.
Branch area to leaf area index (BAILAI) ratio would affect the plant area index (PAI) and the average
scattering phase function of the canopy elements. In our sensitivity analysis the albedo was reduced by
5–16% when the BAILAI increased from 70% to 130% of its default value.
Shoot length would affect the calculation of bidirectional gap probabilities i.e. visibility of the lower
canopy layers and understory. In our sensitivity analysis the effect was negligible: up to 3% increase in
albedo when shoot length increased from 70% to 130% of its default value.
Spectra of leaves/needles, bark, and forest floor would have an effect that is dependent on the relative
share of these scattering components. For some of the species the average contribution of forest floor
on albedo was 50%, and therefore the forest floor would have on average as large an effect as the
leaves/needles/bark have. For most of the species, however, the spectral properties of the tree canopy
dominated.

Literature cited

ASTM Standard G173-03. Standard Tables for Reference Solar Spectral Irradiances: Direct Normal and Hemispherical on a 37 Tilted Surface.

Bird, E. and Riordan, C.: Simple solar spectral model for direct and diffuse irradiance on horizontal and tilted planes at the Earth's surface for cloudless atmospheres. Journal of Climate and Applied Meteorology, 25, 1, 87–97, 1986.

Kuusk, A. and Nilson, T.: A directional multispectral forest reflectance model, Remote Sens. Environ., 72, 244–252, 2000.

Lang, M., Nilson, T., Kuusk, A., Kiviste, A. and Hordo, M.: The performance of foliage mass and crown radius models in forming the input of a forest reflectance model: A test on forest growth sample plots and Landsat 7 ETM+ images, Remote Sens. Environ., 110(4), 445–457, 2007.

Monteith, J. L.: Solar radiation and productivity in tropical ecosystems. Journal of Applied Ecology, 9, 744–766, 1972.

Nilson T. and Peterson U.: A forest canopy reflectance model and a test case. Remote Sens. Environ. 37, 131–142, 1991.

Rautiainen, M., Mõttus, M., Stenberg, P. and Ervasti, S.: Crown envelope shape measurements and models, Silva Fenn., 42(1), 19–33, 2008.

Rautiainen, M., Stenberg, P., Nilson, T. and Kuusk, A.: The effect of crown shape on the reflectance of coniferous stands, Remote Sens. Environ., 89, 41–52, 2004.

Stoy, P. C., Street, L. E., Johnson, A. V, Prieto-Blanco, A. and Ewing, S. A.: Temperature, heat flux, and reflectance of common subarctic mosses and lichens under field conditions: might changes to community composition impact climate-relevent surface fluxes?, Arct. Antarct. Alp. Res., 44(4), 500–508, doi:10.1657/1938-4246-44.4.500, 2012.

Widlowski, J. L., Taberner, M., Pinty, B., Bruniquel-Pinel, V., Disney, M., Fernandes, R., Gastellu-Etchegorry, J. P., Gobron, N., Kuusk, A., Lavergne, T., Leblanc, S., Lewis, P. E., Martin, E., Mõttus, M., North, P. R. J., Qin, W., Robustelli, M., Rochdi, N., Ruiloba, R., Soler, C., Thompson, R., Verhoef, W., Verstraete, M. M. and Xie, D.: Third Radiation Transfer Model Intercomparison (RAMI) exercise: Documenting progress in canopy reflectance models, J. Geophys. Res. Atmos., 112, 1–28, 2007.

List of relevant changes in the manuscript

· Calculated and reported results on albedo and FAPAR in white-sky conditions. Methods described in
Section 2.2.1, results in Section 3.1 and in Tables 4 and 5.

· Calculated albedo and FAPAR in blue-sky conditions in order to test the effect of atmosphere on the
results. The findings are summarized in Section 2.2.1.

· Added a note about LUE concept and reference to Monteith 1972 (Section 1).

· Changed terminology used for incoming and outgoing fluxes (Section 2.2.1).

· Added a note that similarly as for $FAPAR_{TOT}$, also in the case of $FAPAR_{CAN}$ green biomass was not
separated (Section 2.2.1).

· Added discussion on the sensitivity of FRT to its input parameters (Section 2.2.2).

· Added discussion on the role of forest floor, particularly in light of reindeer grazing (Section 4)

· Changed the wordings throughout the text in order to avoid making the impression that too broad
generalizations were made (e.g. in the abstract: "forest in Alaska" changed to "studied plots in Alaska")

· Added a reference to Lukeš et al. 2016, which was published since the submission of our manuscript,
and discussed our results in relation to those obtained in their study (Sections 1 and 4).

[revised manuscript text omitted]

---

## Author Response (AR2)

Comments to the Author:

Dear,

Since your submission of the revised manuscript a month ago, seven external reviewers (including the initial
reviewers) have been contacted for advice but all rejected or missed the deadlines. To avoid further delay, I
decided to base the decision solely on my own judgement.

- 7
- 8 Comments:

Readers should be able to read the manuscript and benefit from the discussion phase without having to go the 10 journal website and actually read the discussion. The assumptions, insights and caveats identified by the 11 reviewers should be used to improve the manuscript. This implies that the responses addressing the reviewers 12 concerns – which satisfied the reviewers and myself- should be integrated in the manuscript.

When providing a revised manuscript, please, include a version of the manuscript where the changes are clearly marked ("Regarding author's changes, a marked-up manuscript version (track changes in Word, latexdiff in LaTeX) converted into a \*.pdf and including the author's response must be provided" copied from http://www.biogeosciences.net/for\_authors/submit\_your\_manuscript.html).

I'm looking forward to receive a revised manuscript at the earliest of your convenience,

- 20
- 21 Sebastiaan Luyssaert
- 22

Dear Dr. Luyssaert,

We have prepared the second revision of our manuscript according to your comments. It is true that some of the issues discussed during the review process were not fully reflected in the manuscript. Particularly, we see now that explanations for some of our choices regarding model parameters or calculations were lacking in the manuscript, although they were included in our response to the reviewers. We hope that the revised version is more thorough in that sense.

In the following table we list the most important concerns raised by you and the reviewers, and explain
 how we modified the manuscript in order to better reflect the discussion related to these concerns.

| Validation and sensitivity of
the FRT model (Reviewer #1,
Editor)                                          | We added a reference to the original Nilson and Peterson (1991) paper,
and a reference to the RAMI website where the reader can view the
results of comparison to other models (Section 2.2.1). Now the literature
list is more complete for those readers that are interested in the model
validation and theoretical background.
We also added an explanation for why we chose not to model different
tree size classes and a short discussion explaining how including several
size classes may affect the simulation results (Section 2.2.2). This
completes the discussion about model sensitivity to the most important
parameters (tree size classes, crown radius, crown shape) that the
reviewer #1 was concerned about. |
|------------------------------------------------------------------------------------------------------------------|-----------------------------------------------------------------------------------------------------------------------------------------------------------------------------------------------------------------------------------------------------------------------------------------------------------------------------------------------------------------------------------------------------------------------------------------------------------------------------------------------------------------------------------------------------------------------------------------------------------------------------------------------------------------------------------------------------------------------------------------------------------------|
| Importance of forest floor
and the correlation between
forest floor and overstory
(Reviewer #1, Editor) | We think that the additions of text into Discussion Section in the first
revision are enough to highlight the importance of forest floor.
Regarding the correlation between overstory and forest floor cover, we
now reported the results observed in our data, and added an
explanation for why spatial variation in the forest floor reflectance was
not modeled (end of Section 2.2.2). Now our reasoning behind the
choices made in modeling forest floor reflectance should be evident for
the reader without reading the online discussion forum.                                                                                                                                                                                    |
| Generality of results
(Reviewer #1)                                                                           | We think that the re-wordings that we implemented in the first revision
round were enough to answer this concern.                                                                                                                                                                                                                                                                                                                                                                                                                                                                                                                                                                                                                                            |
| Inclusion of a diffuse
illumination scenario, the
effect of atmosphere on the
results (Reviewer #2)     | We re-formulated the text in which we describe the black- and white-sky calculations and report the results of our test in which we compare the use of top-of-atmosphere vs. bottom-of-atmosphere solar spectra (last two paragraphs of Section 2.2.1). We also added a couple of sentences that justify to the reader why the atmosphere was ignored. We think that the revised text now well justifies our choices regarding the assumptions on atmosphere and angular properties of incoming solar radiation.                                                                                                                                                                                                                                                |
| Separation of green vs. total
FAPAR (Reviewer #2)                                                             | We added text which explains why green FAPAR was not calculated (Section 2.2.1).                                                                                                                                                                                                                                                                                                                                                                                                                                                                                                                                                                                                                                                                                |
| Modeling light use efficiency
(Reviewer #2)                                                                   | We did not add discussion on this topic. We already state in the introduction that FAPAR is not exactly productivity, although it is commonly used as a proxy of it. Thus, this assumption will be clear to the reader right from the beginning.                                                                                                                                                                                                                                                                                                                                                                                                                                                                                                                |

[revised manuscript text omitted]

, (2)

$$FAPAR_{TOT} = \mathop{a}\limits^{700}_{I=400} w_I \times a_I^T$$
, (3)

The weights  $(w_{\lambda})$  were obtained from the solar irradiance spectrum. Solar irradiance values (W m-2) were scaled by dividing them with the total solar irradiance within the spectral region used (i.e., 400–2100 or 400–700 nm). The weights were thus unitless and summed up to unity. FAPARTOT and FAPARCAN were separated because the former is a measure of total ecosystem productivity whereas the latter is more closely linked with timber production. Our FAPARCAN and FAPARTOT do not separate green biomass from woody or dead branches or from litter on the ground, and the values therefore represent upper limits of available solar energy for photosynthesis in tree canopy, and in the ecosystem as a whole.

The canopy and total absorptions needed for FAPAR determination were obtained using upward scattered ( $f_1$  - ) and downwelling ( $f_1$  - ) fractions of incoming radiation, and the reflectance factor of the forest floor ( $r_g$ ) as follows:

$$a_{I}^{c} = 1 - f_{I} - f_{I}^{c} + f_{I}^{c} \times f_{I}^{c}$$
, (4)

$$a_i^T = 1 - f_i - ,$$
 (5)

[revised manuscript text omitted]
 FAPARCAN. At the lowest SZA (40°) the species-specific FAPARCAN (Table 4) was strongly correlated with species-specific LAIeff (Table 1) (r = 0.93). At large SZAs the canopy interception approached 100% at almost all LAIeff 417 values (cf. Fig. 5c,d) and FAPARCAN was therefore mainly determined by the absorption of the foliage at PAR wavelengths. 418 419 Leaves of broadleaved trees absorbed less than conifer needles, which explains why FAPARCAN of broadleaved species did 420 not increase as rapidly as a function of SZA as did FAPARCAN of coniferous species (Table 4).

**421 3.2 Relation of albedo to FAPAR**

FAPARCAN was negatively correlated with albedo in conifer dominated forests (Fig. 8). The correlation was strongest at the 423 smallest SZA (r = -0.91, r = -0.90) and weakest at the largest SZA (r = -0.63, r = -0.59). When including mixed plots and the 424 plots dominated by broadleaved trees, correlation of FAPARCAN to albedo varied from almost non-existent in Alaska (r 425 ranging from -0.17 to 0.07) to moderate in Finland (r ranging from -0.62 to -0.30). The higher correlation in Finland can be 426 explained by the small number of broadleaved dominated forests in our data from Finland. In addition to the proportion of 427 broadleaved trees, variation in forest floor characteristics influenced the albedo-FAPARCAN relations by altering the albedo values (Fig. 8). The effect of forest floor was seen in relatively sparse canopies only. For example, at SZA of 40° the effect 428 429 of forest floor on albedo started to show at FAPARCAN values below 0.5 (Fig. 8). Remembering that FAPARCAN was tightly 430 related to LAIeff, this value corresponds LAIeff of approx. 1. FAPARTOT 
[revised manuscript text omitted]